

# Gene selection based on adaptive neighborhood-preserving multi-objective particle swarm optimization

Sumet Mehta[1,2], Fei Han[1], Muhammad Sohail[3], Bhekisipho Twala[4], Asad Ullah[3], Fasee Ullah[5], Arfat Ahmad Khan[6] and Qinghua Ling[7]

[1] School of Computer Science & Communication Engineering, Jiangsu University, Zhenjiang, Jiangsu, China
[2] R&D, Star Engineers India Pvt. Ltd., Pune, Maharashtra, India
[3] Department of Computer Software Engineering, Military College of Signals, NUST, Islamabad, Islamabad, Pakistan
[4] Faculty of Information and Communication Technology, Tshwane University of Technology, Pretoria West, Pretoria, South Africa
[5] The Department of Computing, Universiti Teknologi PETRONAS, Seri Iskandar, Perak Darul Ridzuan, Malaysia
[6] Department of Computer Science, College of Computing, Khon Kaen University, Khon Kaen, Khon Kaen, Thailand
[7] School of Computer Science and Engineering, Jiangsu University of Science & Technology, Zhenjiang, Jiangsu, China

Corresponding authors
Fei Han, hanfei@ujs.edu.cn
Bhekisipho Twala,
bhekisiphotwala@gmail.com

## ABSTRACT

The analysis of high-dimensional microarray gene expression data presents critical challenges, including excessive dimensionality, increased computational burden, and sensitivity to random initialization. Traditional optimization algorithms often produce inconsistent and suboptimal results, while failing to preserve local data structures limiting both predictive accuracy and biological interpretability. To address these limitations, this study proposes an adaptive neighborhood-preserving multi-objective particle swarm optimization (ANPMOPSO) framework for gene selection. ANPMOPSO introduces four key innovations: (1) a weighted neighborhood-preserving ensemble embedding (WNPEE) technique for dimensionality reduction that retains local structure; (2) Sobol sequence (SS) initialization to enhance population diversity and convergence stability; (3) a differential evolution (DE)-based adaptive velocity update to dynamically balance exploration and exploitation; and (4) a novel ranking strategy that combines Pareto dominance with neighborhood preservation quality to prioritize biologically meaningful gene subsets. Experimental evaluations on six benchmark microarray datasets and eleven multi-modal test functions (MMFs) demonstrate that ANPMOPSO consistently outperforms state-of-the-art methods. For example, it achieves 100% classification accuracy on Leukemia and Small-Round-Blue-Cell Tumor (SRBCT) using only 3–5 genes, improving accuracy by 5–15% over competitors while reducing gene subsets by 40–60%. Additionally, on MMFs, ANPMOPSO attains superior hypervolume values (*e.g.*, $1.0617 \pm 0.2225$ on MMF1, approximately 10–20% higher than competitors), confirming its robustness in balancing convergence and diversity. Although the method incurs higher training time due to its structural and adaptive components, it achieves a strong trade-off between computational cost and biological relevance, making it a promising tool for high-dimensional gene selection in bioinformatics.

## INTRODUCTION

Microarray technology enables the high-throughput measurement of gene expression levels across thousands of genes simultaneously. This technology has had a transformative impact on multiple domains, ranging from early disease detection and personalized treatments (*DeGroat et al., 2024*) to drug safety and development (*Afshari, Nuwaysir & Barrett, 1999*), guiding dietary and nutritional recommendations (*Warburton et al., 2018*), assessing environmental health impacts (*Espín-Pérez et al., 2018*), and enabling genetic counselling for informed family planning (*Sahoo et al., 2017*). Microarray data is structured as a matrix, where rows represent genes, columns represent samples, and each cell denotes the expression level of a specific gene in a given sample. However, the high dimensionality, high noise levels, and small sample sizes inherent to microarray datasets pose significant challenges in applications such as medical diagnosis and prognosis, where distinguishing between normal and abnormal tissues is of paramount importance (*Han et al., 2015*). Consequently, identifying the most discriminatory genes remains a critical challenge in microarray data analysis.

Gene selection is a key process in addressing these challenges. By identifying a subset of genes most relevant to specific phenotypes or clinical outcomes, gene selection enhances the interpretability, computational efficiency, and predictive performance of microarray analysis (*Wu et al., 2019*; *Fustero-Torre et al., 2021*; *Mehta et al., 2025*). Recent advancements in machine learning and optimization techniques have played a pivotal role in extracting biologically meaningful insights from microarray data, driving substantial progress in the field (*Wu et al., 2023*; *Zhang et al., 2023*). In parallel, the growing field of deep learning and data fusion has further underscored the need for robust feature selection methods. For example, *Zhao et al. (2024)* review deep learning–based cancer data fusion techniques that integrate heterogeneous sources to improve diagnostic accuracy, while *Wang, Li & Ma (2025)* introduced the Multi-Scale Three-Path Network (MSTP-Net) for retinal vessel segmentation, demonstrating the potential of multi-scale feature extraction to capture both local details and global structures.

Among various gene selection methods, evolutionary algorithms have emerged as a powerful approach for handling the computational complexity of high-dimensional data while achieving superior predictive performance. In particular, swarm-based techniques, a subset of bio-inspired meta-heuristics, have gained widespread attention. Swarm intelligence (SI), a subset of artificial intelligence inspired by collective behaviors in distributed and self-organized systems, has been instrumental in this context. Several SI-based approaches, such as particle swarm optimization (PSO) (*Han et al., 2019*), genetic algorithm (GA) (*Li, Wu & Tan, 2008*), simulated annealing (*Filippone, Masulli & Rovetta, 2011*), and biogeography-based optimization (BBO) (*Li & Yin, 2013*), have been applied to gene selection tasks. Among these, PSO has demonstrated particular effectiveness in

addressing the challenges of high-dimensional microarray data. PSO is a bio-inspired, nonlinear optimization algorithm that models the social behaviors observed in bird flocks (*Kennedy & Eberhart, 1995*). In PSO, each particle represents a potential solution, and the algorithm iteratively searches for the global optimum. However, traditional PSO-based gene selection methods typically employ a single-objective function, focusing on guiding the population toward a specific gene set.

Gene selection inherently involves two conflicting objectives: maximizing the relevance of selected genes to the target class and minimizing redundancy among the selected genes (*Wang et al., 2020*). Balancing these objectives requires careful trade-offs, making multi-objective optimization approaches essential for robust and biologically meaningful outcomes. To address the limitations of single-objective optimization, researchers have developed multi-objective particle swarm optimization (MOPSO)-based methods, which optimize multiple conflicting objectives using Pareto-based optimization (*Rahimi et al., 2023*; *Han et al., 2024*). For instance, a hybrid MOPSO algorithm was proposed in *Han et al. (2021)* for gene selection in microarray cancer classification, demonstrating improved performance over traditional single-objective techniques by providing a more comprehensive optimization framework. Similarly, *Xue, Zhang & Browne (2012)* introduced the Crowding, Mutation, and Dominance-based Particle Swarm Optimization for Feature Selection (CMDPSOOFS) algorithm, which incorporated crowding distance, dominance mechanisms, and mutation techniques to enhance convergence and diversity.

Despite significant progress, existing MOPSO-based methods still suffer from notable limitations that hinder their effectiveness. For example, the CMDPSOFS algorithm prioritizes diversity through crowding distance and mutation mechanisms but lacks effective local search strategies, limiting its ability to generate optimal feature subsets. Furthermore, the performance of MOPSO-based methods is highly influenced by the quality of the initial population. Randomly generated initial populations, commonly used in PSO, can lead to challenges such as prolonged convergence times, suboptimal gene set selection, or premature convergence to local optima (*Gao et al., 2024*; *Tijjani, Ab Wahab & Noor, 2024*; *Yang et al., 2024*). A well-structured initial population enhances diversity in solution space exploration, facilitating faster convergence and improved solution quality. Conversely, poorly designed initial populations may undermine algorithm performance and efficiency.

Another major limitation of existing MOPSO-based methods is their inability to effectively preserve local structures within the gene space, which is essential for maintaining diversity and capturing biologically meaningful relationships. For instance, a MOPSO-based feature selection approach introduced in *Xu et al. (2024)* neglects local structural information, reducing its sensitivity to localized effects and increasing redundancy among selected features. Other methods, such as Many-objective Particle Swarm Optimization for Graph-based Gene Selection in Medical Diagnosis Problems (MaPSOGS) (*Azadifar & Ahmadi, 2021*), which integrates a graph-based framework for gene selection, focus on increasing diversity through clustering and repair operators without explicitly maintaining local relationships. Similarly, approaches like Cooperative Coevolutionary Multi-guide Particle Swarm Optimization (CCMGPSO)

(*Madani, Engelbrecht & Ombuki-Berman, 2023*) and Multi-objective Particle Swarm Optimization with Node Centrality-based Feature Selection (MPSONC) (*Rostami et al., 2020*) emphasize global optimization while oversimplifying gene relationships by converting them into graph representations, potentially losing critical neighborhood information.

Moreover, most MOPSO algorithms aim to explore all Pareto-optimal solutions under the assumption that all non-dominated solutions are equally desirable (*Konak, Coit & Smith, 2006*; *Ishibuchi, Tsukamoto & Nojima, 2008*). However, in gene selection, the primary objective is to enhance classification performance, prioritizing regions with higher predictive accuracy rather than merely minimizing the number of selected genes on the Pareto front. Consequently, existing methods often waste computational resources by searching for less relevant solutions (*Deb & Sundar, 2006*; *Thiele et al., 2009*), limiting their ability to fully capture complex gene interdependencies and explore the solution space effectively. In addition, recent approaches such as Multi-objective Feature Selection using Real-valued Encoding and a Preference Leadership Strategy (MOFS-REPLS) (*Fu et al., 2024*), rough hypervolume-driven methods (*Zhou et al., 2025*), and adaptive deep learning techniques (*Li et al., 2020*) have advanced feature selection, they do not fully address local structure preservation or adaptive initialization within a MOPSO context. Therefore, current MOPSO-based gene selection methods suffer from sensitivity to random initialization, neglect of local structural preservation, and imbalanced exploration-exploitation dynamics, limiting their biological relevance and optimization efficacy.

As summarized in Table 1, these limitations persist across state-of-the-art MOPSO-based methods. To overcome these challenges, we propose an adaptive neighborhood-preserving multi-objective particle swarm optimization (ANPMOPSO) framework for gene selection in microarray analysis. This approach addresses the limitations of existing MOPSO-based methods by incorporating adaptive mechanisms to preserve local structures, improving population initialization, and aligning optimization objectives with the practical goals of gene selection. The primary contributions of ANPMOPSO are as follows:

- ANPMOPSO employs weighted neighborhood-preserving ensemble embedding (WNPEE) technique to reduce the dimensionality of microarray data while retaining essential local structural information. The use of Sobol sequence (SS) for population initialization enhances diversity and ensures an efficient exploration of the solution space from the outset.

- Unlike previous methods that primarily focus on gene relevance and redundancy, ANPMOPSO simultaneously optimizes classification accuracy, gene count, and neighborhood preservation quality, providing a holistic evaluation of gene subsets.

- ANPMOPSO integrates a differential evolution (DE)-based adaptive velocity update mechanism to dynamically balance exploration and exploitation. This mechanism maintains diversity in both the decision space and objective space, promoting convergence toward high-quality solutions.

**Table 1 Comparative analysis of MOPSO-based gene selection methods.**

| Method | Initialization | Local structure preservation | Adaptive mechanism | Novel ranking strategy |
|---|---|---|---|---|
| CMDPSOFS (*Xue, Zhang & Browne, 2012*) | Random | × | × (Crowding distance) | × (Dominance-only) |
| MOPSO-ASFS (*Han et al., 2021*) | Random | × | × (Fixed parameters) | × (Adaptive dominance) |
| MaPSOGS (*Azadifar & Ahmadi, 2021*) | Random | ✓ (Graph-based clustering) | × | × (Clustering-guided) |
| MPSONC (*Rostami et al., 2020*) | Random | × | × | ✓ (Node centrality) |
| CCMGPSO (*Madani, Engelbrecht & Ombuki-Berman, 2023*) | Random | × | × | × (Cooperative coevolution) |
| MOFS-REPLS (*Fu et al., 2024*) | ReliefF-guided with roulette wheel sampling | × | ✓ (Real-valued encoding + preference leadership strategy) | ✓ (Preference leadership) |
| RHV-FS (*Zhou et al., 2025*) | Random | ✓ (Rough set theory) | ✓ (Groupwise intelligent sampling) | ✓ (Hypervolume-guided selection) |
| ANPMOPSO (Proposed) | ✓ (Sobol Sequence) | ✓ (WNPEE) | ✓ (DE Mutation) | ✓ (Pareto + Neighborhood preservation) |

- ANPMOPSO introduces a novel selection strategy that combines Pareto dominance with neighborhood preservation quality, prioritizing solutions that not only achieve superior performance across multiple objectives but also retain biologically meaningful local structures.

   By addressing these challenges, ANPMOPSO facilitates the discovery of biologically significant gene subsets, enhances classification performance, and reduces computational overhead, providing a robust and efficient framework for microarray data analysis.

   The remainder of this article is structured as follows: Preliminaries provide the necessary background for understanding the proposed approach. Proposed Method details the ANPMOPSO framework, highlighting its methodology and key innovations. Results and Discussions present experimental findings, demonstrating the effectiveness of ANPMOPSO on multimodal multi-objective test functions (MMFs) and microarray datasets. Finally, Conclusions summarize the key findings and discuss potential future research directions.

## PRELIMINARIES

### Multi-objective optimization problems

Several biomedical enhancement obstacles require synchronized optimization of distinct multiple objectives, known as multi-objective problems (MOPs) (*Kong et al., 2023*; *Peng & Guo, 2023*). MOPs have two main goals: the diversity and convergence of Pareto optimal solutions (POSs). The aim of diversity is to optimize a set of POSs as distinct as possible and convergence is to optimize a set of optimal solutions as close as possible to the Pareto front (*Yang et al., 2020*). Generally, the minimization problem for MOP is articulated as:

*Minimize $Y = f(x) = f_1(x), f_2(x), \ldots, f_m(x)$*        (1)
*Subject to $x \in \Omega$*

where $\Omega$ is the decision space and $Y$ is objective space. $y = (y_1, y_2, \ldots, y_m) \in Y$ is a $m$-dimension objective vector and $x = (x_1, x_2, \ldots, x_D) \in \Omega$ is an $D$-dimension decision variable vector. Several definitions associated with multi-objective optimization are given as follows:

    *Definition 1 (Pareto optimal):* A decision vector $x^* \in X$ is said to be Pareto optimal if there is no other $x \in X$, such that $x \prec x^*$.

    *Definition 2 (Pareto optimal set):* For a given MOPs, the Pareto optimal set is defined as:

$$POS = \{x \in X \mid \nexists z \in X, z \prec x\} \tag{2}$$

    *Definition 3 (Pareto front):* For a given MOPs, the Pareto front set is defined as:

$$PF = \{F(X) \mid x \in PS\} \tag{3}$$

    Further in multi-modal multi-objective (MMO) problems, a "global POS" is a group of solutions where none of them is outperformed by any other solutions in the entire possible range. On the other hand, a "local POS" is a group of solutions where none of them are dominated by their close neighbors. For a multi-objective optimization problem to be considered a MMO problem, it needs to meet one of following two conditions: either possess at least one local POS or have more than one global POS.

    It can be realized from aforementioned that a global POS is distinct from a local. Consequently, if a problem includes a local POS, it necessarily includes at least one global POS. In rare situations, the local POS might consist of only one solution.

## Particle swarm optimization

PSO is a single-objective computational procedure; encouraged by collective actions of the birdies that can be defined as a repeatedly budding system (*Kumar, Pandey & Ahirwal, 2023*; *Li et al., 2023*). It is a community-based speculative procedure that aims to optimize the ideal solution to a specified optimization problem. PSO has been used in various biomedical studies such as gene selection and cancer classification due to its efficacy of fast convergence and simplicity of employment (*Chen et al., 2014*; *Jain, Jain & Jain, 2018*).

    PSO operates through two primary components: the exploration aspect, signified by the inertia weight ($w$) and the cognitive coefficient ($c_1$), dictating an individual's peak position influence; and the exploitation facet, denoted by the social coefficient ($c_2$), governing the effect of the swarm's peak position. PSO implementation mainly includes: initialization of each individual, main loop that contains iterations for updating velocities of these individuals according to peak positions as well as global positions and determine the fitness function, and finally termination. Let $x_i$ and $v_i$ is the current position and current velocity of the particle and $r_1$ and $r_2$ are arbitrary numbers in the range [0, 1]. Then, particles travel conforming to the following equations throughout the exploration process.

$$v_i = wv_x + c_1 r_1 \left(p^{peak}{}_i - x_i\right) + c_2 r_2 \left(g^{peak}{}_x - x_i\right) \tag{4}$$

$$x_i = x_i + v_i \quad 1 \leq i \leq n. \tag{5}$$

By iteratively updating the individual positions based on their velocities, PSO aims to converge towards the optimal solution by balancing exploration and exploitation. The algorithm's success depends on carefully selecting the values for the parameters $(w, c_1, c_2)$ and the appropriate termination criterion for the specific problem at hand. Also, it can be noted that the PSO algorithm is versatile and can be adapted to different optimization problems by customizing the fitness evaluation function and problem-specific constraints.

## Sobol sequence

Sobol sequence (SS) is a type of quasi-random sequence used in numerical computations, particularly in Monte Carlo simulations and optimization algorithms (*Atanassov & Ivanovska, 2022*). Unlike pseudo-random sequences generated by traditional methods, such as the linear congruential generator, SS offers superior properties in terms of low discrepancy and high-dimensional uniformity. Developed by the Russian mathematician Ilya M. Sobol, this sequence is designed to cover the entire domain space more evenly, leading to improved convergence rates and accuracy in numerical integration and optimization tasks (*Hu et al., 2024*; *Zhang et al., 2024*). Sobol Sequence finds applications in various fields, including finance, engineering, and computer graphics, where high-quality random number generation is crucial for achieving reliable and efficient results.

Firstly, the number of dimensions $d$ and the number of points to generate $n$ are chosen. Then, direction numbers are computed and stored for each dimension $j$, denoted as $\{v_{j,i}\}_{i=1}^{\infty}$. For each dimension $j$, it generates the SS $\{s_{j,i}\}_{i=1}^{n}$ using the following formula:

$$s_{j,i} = \frac{g(j)}{2^j} \oplus \frac{g(j+1)}{2^{j+1}} \oplus \ldots \oplus \frac{g(j+m-1)}{2^{j+m-1}} \tag{6}$$

where $g(j)$ is the integer representation of the Gray code of $j$, $\oplus$ represents the bitwise exclusive OR operation, $m$ is the number of bits required to represent $i$ in binary. The generated Sobol points $\{s_{j,i}\}_{i=1}^{n}$ are typically transformed to the unit interval $[0, 1]$ to represent points in the search space. Finally, the Sobol points in the unit interval can be transformed to the actual search space based on the bounds and constraints of the optimization problem. This process ensures that the points in the SS are distributed evenly across the unit hypercube, leading to improved coverage and convergence properties compared to pseudo-random sequences.

## PROPOSED METHOD

We propose an ANPMOPSO framework to address the challenges of microarray gene selection, including high dimensionality, sensitivity to initial population generation, and inadequate consideration of local structural information in conventional methods. The framework integrates several novel components to ensure the selection of biologically significant gene subsets while enhancing computational efficiency and classification performance. Key innovations include WNPEE for dimensionality reduction (*Mehta, Zhan & Shen, 2019*), SS-based initialization for diverse and efficient population generation,

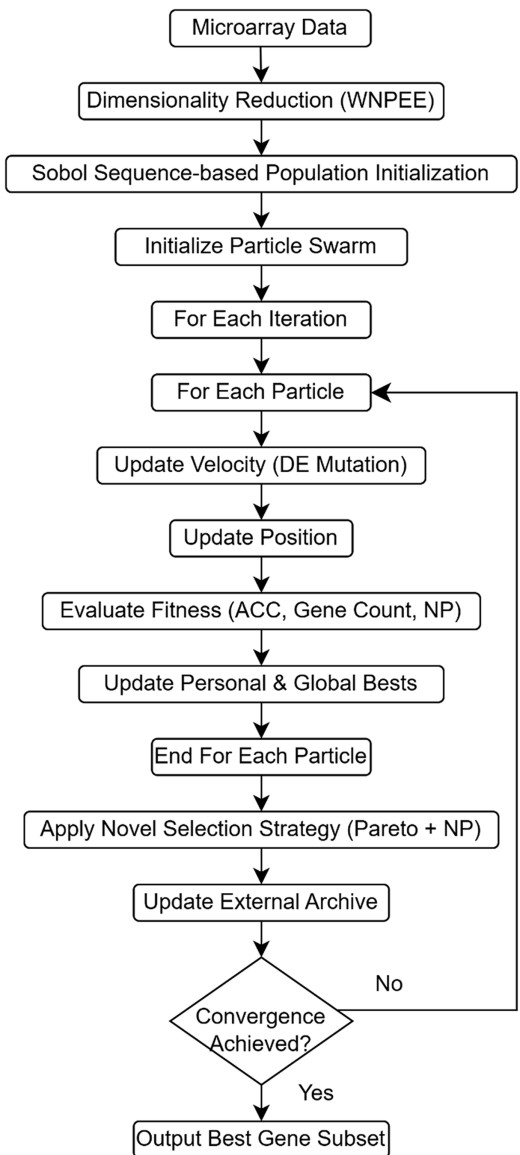

**Figure 1** **The flowchart of the proposed ANPMOPSO gene selection method.**

and an adaptive velocity update mechanism incorporating DE mutation to balance exploration and exploitation.

ANPMOPSO performs multi-objective optimization by simultaneously optimizing classification accuracy, the number of selected genes, and neighborhood preservation quality, ensuring a comprehensive evaluation of candidate gene subsets. A novel ranking mechanism is introduced, integrating Pareto dominance with neighborhood preservation quality, prioritizing solutions that retain meaningful local structures within the data. These advancements collectively overcome key limitations of existing methods, such as random initialization and limited optimization scope, providing a more robust and interpretable framework for microarray analysis. Figure 1 illustrates the overall workflow of the

proposed ANPMOPSO framework. The proposed framework begins with dimensionality reduction, which is described in detail below.

## Dimensionality reduction using WNPEE

To address the challenges of high-dimensional microarray data, we employ WNPEE as a robust dimensionality reduction technique. Unlike traditional methods that rely on a single graph to capture data relationships (*Mehta, Zhan & Shen, 2019*; *Qin et al., 2024*), WNPEE constructs an ensemble of $L$ adjacent graphs $G_1, G_2, \ldots, G_L$, each representing the local structure of the data with varying neighborhood sizes $k$. This ensemble approach enhances neighborhood preservation and improves robustness to parameter sensitivity. The graphs are constructed using the $K$-nearest neighbors (KNN) method, where a directed edge is created between nodes if one node lies within the neighborhood of the other. For each graph, edge weights are computed by minimizing:

$$\min \sum_i \left\| x_i - \sum_j \sum_k \alpha_k W_{k,ij} x_j \right\|^2, s.t \sum_j W_{k,ij} = 1, \; j = 1, 2, \ldots, N \quad (7)$$

where $W_k$ is $N \times N$ weight matrix for each adjacent graph, with $W_{k,ij}$ representing the weight of the edge from node $i$ to $j$, and 0 if no edge exists. The reduced representation $Y$ is obtained by solving the eigenvalue problem:

$$XMX^T\mathbf{a} = \lambda XX^T\mathbf{a} \quad (8)$$

where $M = (I - \sum \alpha_k W_k)^T (I - \sum \alpha_k W_k)$. The transformation matrix $A = [a_0, a_1, \ldots, a_{d-1}]$ maps the high-dimensional data $X$ into a reduced-dimensional space $Y = A^T X$, preserving local structures. The graph weights $\alpha_k$ are iteratively optimized to minimize reconstruction loss, ensuring that the ensemble accurately represents the data's locality, as

$$\alpha_k = \frac{\sqrt[r-1]{1/tr(\Sigma_k)}}{\sum_{j=1}^{L} \sqrt[r-1]{1/tr(\Sigma_j)}} \quad (9)$$

where $r > 1$ is a control parameter for balancing multiple graphs. This iterative process continues until the loss function converges. The detailed pseudocode of WNPEE is presented in Algorithm 1. By incorporating WNPEE, the proposed method ensures that the optimization process operates on a compact and biologically meaningful representation of the data, improving classification performance and reducing computational overhead.

## Initialization

Following pre-processing, the population of gene subsets is initialized using an SS as outlined in previous section. This approach provides a more efficient and structured initialization strategy compared to conventional random methods. Each individual in the population, denoted as $Y_i$, is generated as follows:

$$X_i = X_{min} + s_{j,i} \times (X_{max} - X_{min}) \quad (10)$$

---

**Algorithm 1** WNPEE algorithm.

**Input:** High dimensional data $X$

**Output:** projection vector **a**

**Parameter:** $\alpha_k$

**Initialize:**

- Construct an ensemble of adjacent graphs using $KNN$ with $\alpha_k$ weights

**while** loss not converged **do**

1. Computing the weights on edges with fix $\alpha_k$ acc. to Eq. (7)

2. Obtain projection vector **a** acc. to Eq. (8)

3. Fix **a** and update $\alpha_k$ acc.to Eq. (9)

4. Compute current loss

**End while**

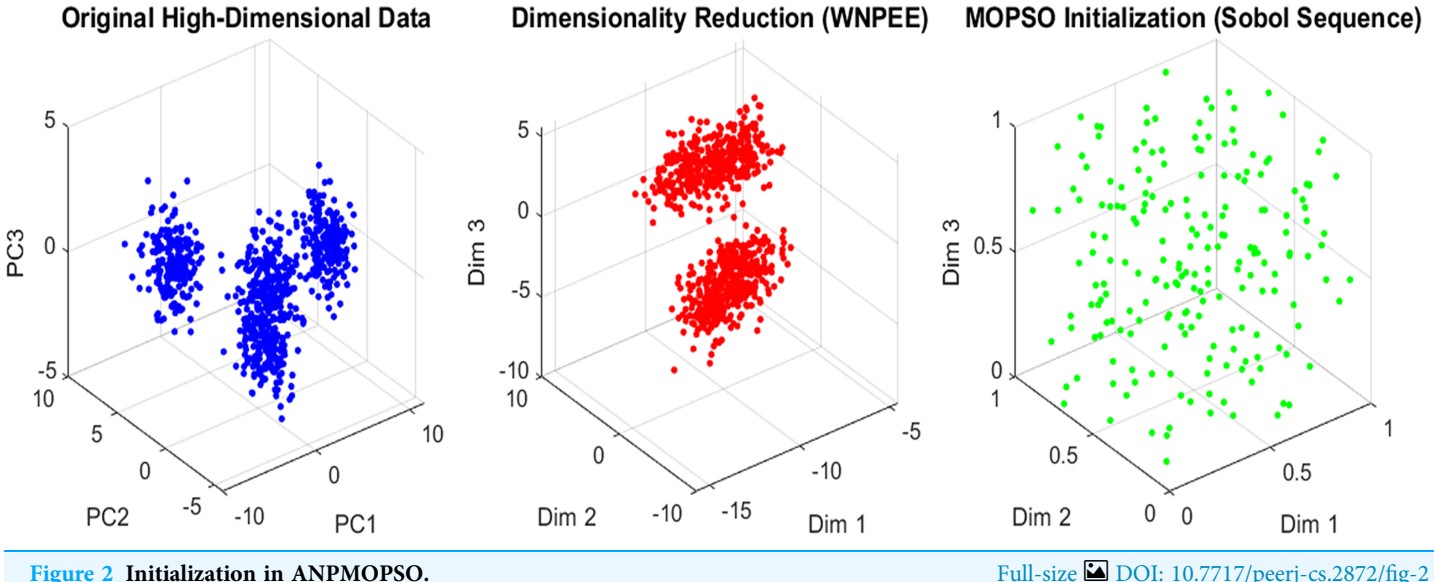

**Figure 2 Initialization in ANPMOPSO.**           

where $X_{max}$ and $X_{min}$ are the upper and lower bounds, respectively. The parameter $s_{j,i}$ represents the *ith* random number generated by the SS as calculated using Eq. (6) within the interval [0, 1]. From the generated initial population, the current optimal gene subset is selected by evaluating fitness values, ensuring that only high-quality solutions proceed. SS enhances the systematic exploration of the solution space, leading to faster convergence and improved overall performance of the MOPSO optimization algorithm. An overview of the initialization scheme in the proposed ANPMOPSO is shown in Fig. 2.

## Fitness evaluation

Once the population is initialized, each particle's solution is evaluated based on multiple objectives, including classification accuracy, gene count, and neighborhood preservation

quality. These objectives provide a comprehensive assessment of gene subsets, ensuring that selected genes maintain predictive performance while capturing relevant biological insights. In ANPMOPSO, we calculate the classification accuracy $CA(X_i)$ as the ratio of correctly classified instances to the total number of instances in the validation set.

$$CA(X_i) = \frac{Number\ of\ correctly\ classified\ instances}{Total\ number\ of\ instances} \times 100. \tag{11}$$

The number of selected genes $NF(X_i)$, in a gene subset can be simply measured by counting the number of genes with a value of 1 in the binary representation of the subset. Let's denote $X_i$ as a binary vector of length $L$, where $L$ is the total number of genes in the dataset. Each element $x_{ij}$ of $X_i$ represents whether gene $j$ is selected or not. $x_{ij} = 1$ if gene $j$ is selected in subset $X_i$, and $x_{ij} = 0$ otherwise. The number of selected genes $NF(X_i)$ in subset $X_i$ can be calculated as:

$$NF(X_i) = \sum_{j=1}^{L} x_{ij}. \tag{12}$$

Let $NP(X_i)$ represent the neighborhood preservation quality of solution $X_i$ and we compute it as the average preservation score across all data points:

$$NP(X_i) = \frac{1}{N} \sum_{i=1}^{N} \frac{1}{k} \sum_{j=1}^{k} ||x_{ij} - y_{ij}|| \tag{13}$$

where $N$ is the total number of data points, $k$ is the number of nearest neighbors considered, $x_{ij}$ and $y_{ij}$ are the $j^{th}$ nearest neighbor of data point $i$ in the original and embedded spaces, respectively, and $||\cdot||$ denotes Euclidean distance (ED), measuring the difference between corresponding neighbors. Therefore, this metric quantifies how well the local neighborhood structure is preserved. A lower value of $NP(X_i)$ indicates better preservation quality, meaning that the local structure is more faithfully retained in the embedded space. The overall fitness of each particle is computed based on the three defined objectives, with equal weights, as:

$$Fitness\ X_i = \{CA(X_i), NF(X_i), NP(X_i)\}. \tag{14}$$

## Optimization

The optimization loop of ANPMOPSO employs MOPSO with DE-mutation to iteratively refining gene subsets. Each particle in the population maintains a set of solutions representing the Pareto front, which is a set of non-dominated solutions in the objective space. During each iteration, particles dynamically adjust their velocity using DE-mutation inspired mechanisms, facilitating diversity and exploration of the solution space. Particle positions are then updated based on their personal best and the global best found by the entire swarm. The fitness of each particle is evaluated based on multiple objectives, including classification accuracy, gene count, and neighborhood preservation quality. By concurrently optimizing these competing objectives, ANPMOPSO aims to identify diverse, high-quality gene subsets, offering meaningful trade-offs between predictive performance,

subset compactness, and local structure preservation. The update of particle velocities in ANPMOPSO is governed as:

$$v_{i,d}(t+1) = wv_{i,d}(t) + c_1 r_1 \big(pbest_{i,d} - x_{i,d}(t)\big) + c_2 r_2 \big(gbest_d - x_{i,d}(t)\big) + DE\_v_{i,d} \qquad (15)$$

where $v_{i,d}(t)$ represents the velocity of particle $i$ in dimension $d$ at time $t$, $w$ is the inertia weight controlling the impact of the previous velocity, $c_1$ and $c_2$ are acceleration coefficients representing the cognitive and social components, respectively, $r_1$ and $r_2$ are random values sampled from a uniform distribution in the range [0, 1]. $pbest_{i,d}$ is the personal best position of particle $i$ in dimension $d$, $gbest_d$ is the global best position found by the swarm in dimension $d$, and $DE_{v_{i,d}}$ represents the velocity update obtained from the DE-mutation based adaptive mechanism.

The DE-mutation based adaptive velocity update is computed by randomly selecting three other particles $X_{r1}, X_{r2}$, and $X_{r3}$ from the population for each particle $X_i$. A mutant vector $V_{mut}$ is generated by perturbing the quartet using DE-mutation strategy (DE/rand/1) as:

$$V_{mut} = X_{r1} + F \times (X_{r2} - X_{r3}). \qquad (16)$$

This mutant vector is then used to update the particle's velocity adaptively:

$$DE_{v_{i,d}} = K \times (V_{mut} - x_{i,d}(t)). \qquad (17)$$

Here, $F$ is the differential weight factor, $K$ is an adaptive scaling factor determined based on the fitness of the quartet, and $x_{i,d}(t)$ is the current position of particle $i$ in dimension $d$ at time $t$. This adaptive adjustment ensures exploration in regions with potentially better solutions, preventing premature convergence and enhancing diversity. Once the adaptive velocity is obtained, the particle's position is updated as:

$$x_{i,d}(t+1) = x_{i,d}(t) + v_{i,d}(t+1). \qquad (18)$$

Additionally, we ensure that position components are within minimum and maximum bounds $min_{pos}$ and $max_{pos}$, respectively. Further, we evaluate each particle's fitness based on the defined objectives, maximizing classification accuracy, minimizing the number of selected genes, and maximizing neighborhood preservation in the embedded space. This approach allows for the capture of a diverse set of trade-off solutions representing various combinations of classification accuracy, gene count, and neighborhood preservation quality.

## A novel selection strategy

After updating the particles' velocity and position, the newly and previous generated particles are gathered into an external archive. However, given the limited archive size and the necessity to uphold particle quality, it is crucial to implement a suitable selection strategy for archive updates. This selection strategy, employed within the archive, directs the particles search towards the genuine Pareto front, guaranteeing that only the most promising solutions are preserved for further exploration.

Therefore, we propose a novel selection approach that considers both the proximity to the Pareto front and the quality of neighborhood preservation, contrasting with traditional

MOPSO methods that focus solely on proximity. This approach evaluates each solution based on its harmonic mean distance (HMD) to the Pareto front, as opposed to ED used in existing techniques, and its neighborhood preservation quality. By integrating this selection criterion, our method ensures to prioritize solutions that not only achieve good trade-offs among multiple objectives but also preserve meaningful local structures within the microarray data. Therefore, to define a novel selection strategy, we aim to balance two aspects:

- *Proximity to the Pareto front*: In contrast to ED in traditional MOPSOs, which considers only the geometric distance between points in the objective space, the HMD accounts for both the distance to the Pareto front and the diversity of solutions across multiple objectives. By incorporating HMD, our approach effectively diversifies the selection process, ensuring that a broader set of trade-off solutions is explored. Let $HMD(X_i)$ represent the average HMD of solution $X_i$ to the Pareto front, it can be calculated as:

$$HMD(X_i) = \frac{n}{\sum_{j=1}^{n} \frac{1}{d(X_i, X_j)}}. \tag{19}$$

- *Neighborhood preservation quality*: Solutions with better neighborhood preservation in the embedded space are preferred. We compute the neighborhood preservation quality $NP(X_i)$ for solution $X_i$ as the average preservation score across all data points using Eq. (15).

Finally, we define a combined score $S(X_i)$ for each solution $X_i$ as a weighted sum of its HMD to the Pareto front and its neighborhood preservation quality:

$$S(X_i) = \beta \times HMD(X_i) + (1 - \beta) \times NP(X_i) \tag{20}$$

where $\beta$ controls the relative importance of $HMD(X_i)$ and $NP(X_i)$ in the optimization process, higher value of $\beta$ prioritizes $HMD(X_i)$, while a lower value prioritizes $NP(X_i)$. Solutions with lower $S(X_i)$ values are preferred, as they provide a better balance between proximity to the Pareto front and local structure preservation. By integrating this novel selection strategy, our method can effectively identify high-quality solutions that achieve a balance between the multiple objectives of the gene selection while capturing relevant biological insights encoded in the microarray data. Algorithm 2 presents the detailed pseudocode for proposed ANPMOPSO.

## RESULTS AND DISCUSSIONS

### Dataset

To evaluate the performance of the proposed ANPMOPSO framework, we conducted experiments on six publicly available microarray datasets: Brain Cancer, Colon, Leukemia, Lung, Lymphoma, and Small-Round-Blue-Cell Tumor (SRBCT). These datasets were obtained from two primary sources: http://csse.szu.edu.cn/staff/zhuzx/Datasets.Html, and https://github.com/Pengeace/MGRFE-GaRFE. A detailed description of these datasets is presented in Table 2.

---

**Algorithm 2** ANPMOPSO algorithm.

**Input:** Microarray data, Parameters: Population size ($N$), Maximum number of iterations ($max_{iter}$), Number of objectives ($num_{objectives}$), Neighborhood Size ($k$), PSO parameters: inertia weight ($w$), cognitive coefficient ($c_1$), social coefficient ($c_2$), Weighting factor: ($\alpha$)

**Output:** Selected genes set

1. Embed microarray data into a lower-dimensional space using WNPEE using Eq. (8)

2. Initialize population $P$ of $N$ gene subsets using SS Eq. (10)

3. **for** $iter = 1$ $to$ $max_{iter}$ **do**

4.     **for** each particle in population $P$ **do**

5.         Update particle velocity *via* DE mutation $\left(DE_{v_{i,d}}\right)$ using Eq. (17)

6.         Update particle position using **(18)**

7.         Evaluate fitness of particle using objectives: ($CA$, $NG$, $NP$) using Eqs. (11), (12), (13)

8.         Update personal best and global best if necessary

10.     **end for**

11.     Sort the solutions based on a combined score $S(X_i)$ for each solution $X_i$ using Eq. (20)

12.     Solutions with lower values of $S(X_i)$ are considered more favourable and prioritized for selection.

13. **end for**

14. Return selected genes set based on the best solution found

---

**Table 2 Microarray data description.**

| Data | Samples | For training | For testing | No. of classes | Genes |
|---|---|---|---|---|---|
| Brain cancer | 50 | 30 | 20 | 4 | 10367 |
| Colon | 62 | 40 | 22 | 2 | 2000 |
| Leukemia | 72 | 38 | 34 | 2 | 7129 |
| Lung Cancer | 203 | 103 | 100 | 5 | 3312 |
| Lymphoma | 58 | 29 | 29 | 2 | 7129 |
| SRBCT | 83 | 63 | 20 | 4 | 2308 |

These datasets are commonly used for gene selection and classification tasks due to their high dimensionality and biological relevance. All datasets were preprocessed to remove missing values and normalized using min-max scaling to ensure consistency across experiments.

## Hyperparameters

In our experiments, we set the key hyperparameters as:

- Swarm size: 200 particles (optimized for diversity and computational efficiency).
- Iterations: 50 generations (evaluated 30–100 generations; 50 iterations ensured convergence where error rate stabilization ±1% beyond 50).
- Sobol sequence initialization: Dimensions matched the gene count of each dataset, with quasi-random numbers generated in the range [0, 1].
- DE mutation rate ($F$): 0.5 (selected *via* grid search).

- Neighborhood size ($k$): 10 and parameter $\alpha$: 0.5 (optimized for local structure preservation in WNPEE).
- Inertia weight ($w$): 0.7298 (standard PSO configuration).
- Acceleration coefficients ($c_1, c_2$): 1.49445 (empirically validated for balance).
- Parameter ($\beta$): 0.5 (*via* optimally balances HMD and NP for the lowest 1/PSP values).

Other parameters were configured according to corresponding references (*Rostami et al., 2020*; *Shukla, Singh & Vardhan, 2020*; *Azadifar & Ahmadi, 2021*; *Madani, Engelbrecht & Ombuki-Berman, 2023*). To ensure fair comparison and reliable evaluation, these experimental settings were applied consistently across all datasets.

The effectiveness of ANPMOPSO was benchmarked against state-of-the-art metaheuristic algorithms, including MaPSOGS (*Azadifar & Ahmadi, 2021*), Teaching–Learning-Based Optimization and Gravitational Search Algorithm for Feature Selection (TLBOGSA) (*Shukla, Singh & Vardhan, 2020*), CCMGPSO (*Madani, Engelbrecht & Ombuki-Berman, 2023*), and MPSONC (*Rostami et al., 2020*). All experiments were conducted on a desktop system equipped with an Intel Core i7 processor (2.4 GHz) and 16 GB RAM, utilizing MATLAB 2018a (The MathWorks, Natick, MA, USA) as the development environment.

## Performance metrics

To comprehensively evaluate the performance of the proposed ANPMOPSO framework, the following metrics were used:

- *1/Hypervolume (1/HV):* This metric is used to evaluate the convergence and diversity of the Pareto front generated by ANPMOPSO. The hypervolume (HV) is calculated as:

$$HV = vol(\cup_{x \in PF}[x, r]) \tag{21}$$

  where PF is the set of non-dominated Pareto solutions and $r$ is the reference point in objective space. The inverse is used for minimization. Lower values of 1/HV indicate better convergence and spread of the Pareto front.

- *Classification accuracy (ACC):* Classification accuracy is used to quantify the predictive performance of the selected gene subsets, defined in Eq. (11).

- *Error rate:* The error rate is the complement of accuracy and is used to monitor the algorithm's convergence during optimization:

$$Error\ rate = 1 - Acc \tag{22}$$

- *Neighborhood preservation score (NP):* NP evaluates the proportion of nearest neighbors in the original space that remain neighbors in the reduced space same as Eq. (13). A higher NP score indicates better local structure retention.

- *1/PSP (Pareto Sets Proximity):* Pareto Sets Proximity (PSP) quantifies the proximity of obtained Pareto solutions to the true or ideal Pareto front. The PSP is calculated as:

$$PSP = \frac{1}{|P|} \sum_{x \in P} \min_{y \in PF^*} \| x - y \|^2 \tag{23}$$

where $P$ is the set of obtained Pareto solutions, and $PF^*$ is the true Pareto front or its approximation. The lower the PSP, the closer the solutions are to the optimal front. We use 1/PSP to maintain consistency with our minimization objective.

- *Average neighborhood overlap (ANO):* ANO measures how well the local neighborhood of each point is retained after dimensionality reduction.

$$ANO = \frac{1}{N} \sum_{i=1}^{N} \frac{\left| N_k^H(i) \cap N_k^L(i) \right|}{k} \tag{24}$$

where: $N_k^H(i)$ and $N_k^L(i)$ are sets of $k$-nearest neighbors of point $i$ in the high-dimensional space and in the low-dimensional space, and $\left| N_k^H(i) \cap N_k^L(i) \right|$ is the number of overlapping neighbors between the original and reduced feature space. A higher ANO score suggests better neighborhood overlap, implying that dimensionality reduction minimally distorts the original local structure.

- *Spearman's rank correlation (SRC):* SRC evaluates whether the relative ordering of pairwise distances between data points is preserved after dimensionality reduction. It is calculated as:

$$SRC = 1 - \frac{6 \sum_{i=1}^{N} d_i^2}{N(N^2 - 1)} \tag{25}$$

where $d_i$ is difference in rank for point $i$ before and after dimensionality reduction and $N$ is the total number of data points. A higher SRC (close to 1.0) suggests that the relative ranking of distances between points is well-preserved in the reduced space.

These metrics collectively assess not only classification performance but also solution quality, biological relevance, and structural preservation.

## Comparative analysis of hypervolume values

Table 3 presents the mean ± standard deviation of 1/hypervolume (1/HV) values obtained by five multi-objective optimization algorithms: ANPMOPSO, Multi-objective Particle Swarm Optimization with Adaptive Strategies for Feature Selection (MOPSO-ASFS) (*Han et al., 2021*), Omni-optimizer (*Deb & Tiwari, 2005*), Decision Space Niching-based Non-dominated Sorting Genetic Algorithm II (DN-NSGAII) (*Li, Wu & Tan, 2023*), and Multi-objective Particle Swarm Optimization with Ring Topology for Solving Multimodal Multi-objective Problems (MO_Ring_PSO_SCD) (*Yue, Qu & Liang, 2017*), across eleven benchmark test functions. These test problems are derived from *Yue, Qu & Liang (2017)*, and all algorithms are evaluated under identical experimental conditions. Each experiment is repeated 15 times for robustness, and the performance is assessed using the 1/HV performance indicator, where a lower value indicates superior performance. The population size is determined as $100 \times N_{var}$, and the maximum fitness evaluations are set at $4{,}000 \times N_{var}$. In the ANPMOPSO algorithm, the $F$ parameter in Eq. (16) is set to 0.5, and the crossover rate is also 0.5.

The results from Table 3 demonstrate that ANPMOPSO consistently outperforms other optimization algorithms across most benchmark functions, with the exception of MMF7

**Table 3 Mean ± Std 1/Hv values obtained by all compared algorithms on eleven MMFs test functions.**

|  | ANPMOPSO | MOPSO-ASFS | Omni-optimizer | DN-NSGAII | MO_Ring_PSO_SCD |
|---|---|---|---|---|---|
| MMF1 | **1.0617 ± 0.22258** | 1.1445 ± 0.1114 | 1.1437 ± 0.1117 | 1.1445 ± 0.1113 | 1.1466 ± 0.1115 |
| MMF2 | **1.0991 ± 0.42594** | 1.1645 ± 0.1147 | 1.1547 ± 0.1216 | 1.1670 ± 0.1164 | 1.1548 ± 0.1148 |
| MMF4 | **1.7235 ± 0.61687** | 1.7462 ± 0.1118 | 1.7489 ± 0.1115 | 1.7466 ± 0.1168 | 1.7550 ± 0.1122 |
| MMF5 | **1.1419 ± 0.42283** | 1.1436 ± 0.1113 | 1.1444 ± 0.1118 | 1.1438 ± 0.1117 | 1.1466 ± 0.1114 |
| MMF6 | **1.1451 ± 0.1125** | 1.1512 ± 0.1115 | 1.1487 ± 0.1123 | 1.1475 ± 0.1116 | 1.1532 ± 0.1145 |
| MMF7 | 1.1413 ± 8.31198 | **1.1412 ± 0.9383** | 1.1425 ± 0.1115 | 1.1423 ± 0.1113 | 1.1435 ± 0.1114 |
| MMF8 | **2.3660 ± 0.05931** | 2.3854 ± 0.0933 | 2.3790 ± 0.0374 | 2.3832 ± 0.0242 | 2.3727 ± 0.0613 |
| SYM-PART simple | 0.75 ± 5.25e−04 | 0.75 ± 8.66e−05 | 0.75 ± 9.21e−04 | 0.75 ± 1.94e−04 | **0.75 ± 1.65e−03** |
| SYM-PART rotated | **0.74 ± 1.09e−02** | 0.75 ± 9.68e−04 | 0.74 ± 2.55e−04 | 0.75 ± 4.49e−04 | 0.74 ± 3.55e−03 |
| Omni-test ($n = 3$) | **0.0161 ± 3.95e−04** | 0.0161 ± 5.07e−03 | 0.0162 ± 2.46e−04 | 0.0161 ± 4.09e−02 | 0.0162 ± 4.14e−01 |

**Note:**
The bold values indicate the best (lowest) mean 1/Hv performance for each test function.

and SYM-PART simple. In particular, ANPMOPSO achieves the lowest 1/HV values for MMF1, MMF2, MMF4, MMF5, MMF6, and MMF8, highlighting its capability in solving complex multi-objective problems efficiently. Notably, on MMF8, it significantly outperforms other methods (2.3660 ± 0.05931), reinforcing its effectiveness in handling challenging problem landscapes. This consistent superiority suggests that ANPMOPSO is well-suited for diverse optimization scenarios, particularly those requiring robust exploration and exploitation strategies.

However, ANPMOPSO falls slightly behind on MMF7 and SYM-PART simple functions, where MOPSO-ASFS performs slightly better on MMF7 (1.1412 ± 0.9383), and MO_Ring_PSO_SCD marginally outperforms others on SYM-PART simple (0.75 ± 1.65e−03). The close results in SYM-PART functions indicate that these problems do not strongly differentiate algorithm performance. Nonetheless, ANPMOPSO's dominance in most other functions demonstrates its ability to generate high-quality Pareto-optimal solutions and effectively handle multimodal and rotationally symmetric optimization problems. This underscores its robustness and adaptability in solving diverse multi-objective optimization challenges.

## Comparison of true and obtained Pareto-optimal solutions

The true and obtained Pareto-optimal solutions (POSs) for the proposed ANPMOPSO algorithm across all eleven MMFs are shown in Figs. 3A–3K. The results illustrate how well the ANPMOPSO algorithm approximates the true Pareto fronts. In most cases, the obtained POSs (red circles) closely follow the true POSs (blue diamonds), indicating that ANPMOPSO effectively converges toward optimal solutions. This is particularly evident in MMF4, MMF5, and MMF6, where the alignment is nearly perfect, demonstrating the algorithm's capability in solving multimodal and rotationally symmetric functions. However, some variations are observed in MMF1, MMF7, and MMF9, where minor deviations suggest challenges in fully capturing highly complex or oscillatory POSs.

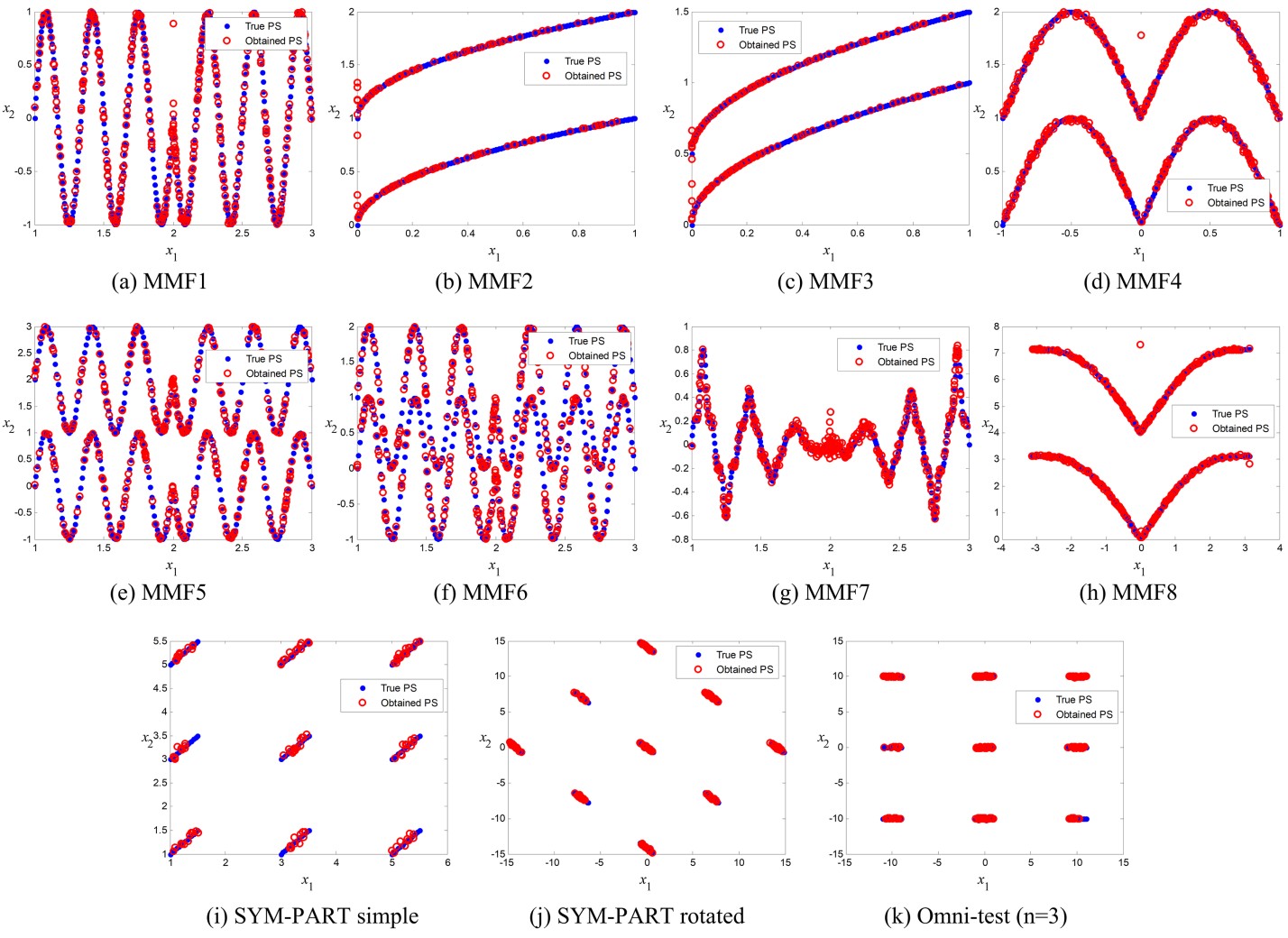

**Figure 3** **Pareto-optimal solutions obtained by ANPMOPSO (using non-dominated sorting and neighborhood preservation scores) on eleven multi-modal multi-objective test functions (MMF1–MMF8, SYM-PART simple, SYM-PART rotated, and Omni-test ($n = 3$)).** Red hollow circles denote ANPMOPSO's solutions; blue solid circles indicate the true Pareto front.

Despite these minor discrepancies, the overall strong agreement between the true and obtained POSs confirms ANPMOPSO's effectiveness in handling diverse multi-objective problems. The observed differences in certain MMFs suggest that additional refinements in exploration strategies or diversity maintenance could further enhance performance. Nonetheless, these results validate the robustness of ANPMOPSO in maintaining convergence and solution accuracy, making it a competitive approach for solving multi-objective optimization problems.

## Comprehensive gene selection comparison

To evaluate the predictive performance of the selected gene subsets, the extreme learning machine (ELM) classifier was used across six microarray datasets. Each experiment was repeated 50 times, and the mean classification accuracies were recorded to ensure the

**Table 4  Classification accuracies (Mean% ± Std) of ANPMOPSO using ELM classifier.**

| Dataset | Test Acc | 5-FOLD CV Acc | LOOCV Acc | Genes selected |
|---|---|---|---|---|
| Brain cancer | 81.27 ± 2.25 | 90.13 ± 1.85 | 91.10 ± 0.58 | 6561, 4927, 5921, 4423 |
| | 81.47 ± 2.69 | 90.07 ± 1.99 | 90.13 ± 1.16 | 3682, 3041, 978, 3051, 4786 |
| | 85.41 ± 3.38 | 90.67 ± 2.53 | 89.98 ± 2.10 | 4433, 7139, 798, 4628, 7045 |
| | 89.77 ± 3.31 | 90.88 ± 2.23 | 92.17 ± 1.44 | 4891, 3051, 1977, 7139, 2891, 885, 2945 |
| Colon | 93.14 ± 2.41 | 96.18 ± 1.32 | 97.83 ± 1.37 | 655, 1989, 1957, 1229, 1080, 1345, 1980, 1872, 1937, 16, 197, 221 |
| | 94.67 ± 3.39 | 95.33 ± 1.52 | 96.17 ± 1.15 | 775, 367, 1760, 16, 197, 271, 165, 59, 1283, 201, 782, 1986, 1120 |
| | 94.92 ± 3.74 | 95.18 ± 1.59 | 96.06 ± 1.11 | 1986, 1121, 1999, 1199, 16, 1599, 102, 108, 782, 1433, 1926, 175 |
| | 95.12 ± 3.86 | 96.06 ± 1.64 | 97.74 ± 1.12 | 1999, 782, 1534, 16, 1986, 1121, 48, 59, 1283, 775, 175, 1770, 271 |
| Leukemia | 100 ± 0.00 | 100 ± 0.00 | 100 ± 0.00 | 2111, 2632, 4060 |
| | 100 ± 0.00 | 100 ± 0.00 | 100 ± 0.00 | 1892, 2632, 4060 |
| | 100 ± 0.00 | 100 ± 0.00 | 100 ± 0.00 | 3268, 4060, 2642 |
| | 94.32 ± 1.32 | 99.08 ± 0.66 | 99.90 ± 0.51 | 1883, 2632, 4233, 4060 |
| Lung | 92.20 ± 1.11 | 98.19 ± 0.72 | 99.25 ± 0.50 | 2871, 1795, 1640, 2789, 2055, 2713, 2867, 545, 1494, 3181 |
| | 91.37 ± 1.99 | 97.12 ± 0.46 | 98.55 ± 0.40 | 609, 866, 369, 3289, 2713, 782, 545, 1494, 570, 2168, 1227 |
| | 91.65 ± 1.26 | 97.33 ± 0.78 | 98.44 ± 0.63 | 2998, 2055, 834, 782, 570, 857, 1795, 881, 2713, 975, 3289 |
| | 91.18 ± 0.90 | 97.15 ± 0.50 | 98.71 ± 0.38 | 975, 2055, 834, 1795, 1484, 2573, 857, 928, 545, 2871, 2713 |
| Lymphoma | 82.62 ± 1.70 | 87.15 ± 2.44 | 89.14 ± 1.32 | 1875, 5873, 2660, 162, 2357, 5689, 458 |
| | 80.52 ± 2.40 | 86.60 ± 2.75 | 88.17 ± 1.81 | 162, 2537, 1875, 550, 1109, 2838, 816 |
| | 81.38 ± 3.10 | 89.07 ± 2.44 | 90.03 ± 1.11 | 5289, 4950, 5459, 1865, 4529, 4697, 1143 |
| | 81.79 ± 1.24 | 89.95 ± 2.33 | 90.29 ± 2.03 | 2838, 2518, 2457, 4839, 6451, 162, 816 |
| SRBCT | 100 ± 0.00 | 100 ± 0.00 | 100 ± 0.00 | 1964, 1013, 440, 2060, 772, 133 |
| | 100 ± 0.00 | 100 ± 0.00 | 100 ± 0.00 | 1975, 565, 265, 1454, 519, 991 |
| | 100 ± 0.00 | 100 ± 0.00 | 100 ± 0.00 | 1921, 2060, 565, 133, 193, 1013, 1499 |
| | 100 ± 0.00 | 100 ± 0.00 | 100 ± 0.00 | 1499, 2055, 2060, 565, 2134, 1964, 133 |

statistical robustness of the results. As shown in Table 4, ANPMOPSO consistently achieved high classification accuracy, with notable performance in Leukemia and SRBCT datasets, where it attained 100% accuracy across all evaluation metrics. This demonstrates the algorithm's capability in identifying the most discriminative genes, leading to highly accurate predictions with a minimal subset of features.

For other datasets such as Brain Cancer, Colon, Lung, and Lymphoma, the proposed method maintained competitive classification performance, with test accuracies frequently exceeding 90%. The five-fold cross validation (CV) and Leave-One-Out Cross-Validation (LOOCV) accuracies further confirm the generalization ability of ANPMOPSO, as they closely align with test accuracy values, minimizing concerns about overfitting. The variation in selected gene subsets across datasets highlights the adaptability of the algorithm in capturing dataset-specific patterns. While minor fluctuations in accuracy were observed in some cases, the overall performance indicates that ANPMOPSO is capable of effectively handling high-dimensional biological data while maintaining robust predictive accuracy. These findings validate the effectiveness of ANPMOPSO as a gene

**Table 5 The Top 10 repeatedly selected genes by ANPMOPSO using ELM classifier.**

| Dataset | Gene no. | Gene name | Description |
|---|---|---|---|
| Brain cancer | 18 | AB000895 | Dachsous 1 (Drosophila) |
| | 4413 | U39817 | Bloom syndrome |
| | 4502 | H78537 | ADAM metallopeptidase domain 12 (meltrin alpha) |
| | 3041 | M64934 | Kell blood group |
| | 7129 | Z97074 | Rab9 effector protein with kelch motifs |
| | 2881 | M57506 | Chemokine (C-C motif) ligand 1 |
| | 2234 | W04668 | ATPase family, AAA domain containing 2 |
| | 6732 | Y00317 | UDP glucuronosyltransferase 2 family, polypeptide B4 |
| | **5081** | **N70358** | **Growth hormone receptor** |
| | 4657 | U51095 | Caudal type homeo box transcription factor 1 |
| Colon | 14 | H20709 | Myosin light chain alkali, smooth-muscle isoform (Human) R |
| | 237 | T50334 | 14-3-3-like protein GF14 omega (*Arabidopsis thaliana*) |
| | 1482 | T64012 | Acetylcholine receptor protein, delta chain precursor (Xenopuslaevis) |
| | 1635 | M36634 | Human vasoactive intestinal peptide (VIP) mRNA, complete cds |
| | 698 | T51261 | Glia Derived Nexin Precursor (Musmuscu-lus) |
| | 141 | D21261 | Sm22-alpha homolog (Human) |
| | 792 | R88740 | Atp synthase coupling factor 6, mitochondrial precursor (Human) = |
| | 3 | R39465 | Eukaryotic initiation factor 4A (*Oryctolagus cuniculus*) |
| | 251 | U37012 | Human cleavage and polyadenylation specificity factor mRNA, complete cds |
| | 23 | R22197 | 60S ribosomal protein L32 (Human) R |
| Leukemia | 4050 | X03934 | GB DEF = T-cell antigen receptor gene T3-delta |
| | 4847 | X95735 | Zyxin |
| | **2671** | **N03128** | **Spectrin, beta, non-erythrocytic 1** |
| | 6567 | T67821 | Acidic (leucine-rich) nuclear phosphoprotein 32 family, member |
| | 1882 | M27891 | CST3 Cystatin C (amyloid angiopathy and cerebral hemorrhage) |
| | 2642 | U05259 | MB-1 gene |
| | 2121 | M63138 | CTSD Cathepsin D (lysosomal aspartyl protease) |
| | **1294** | **L13852** | **Ubiquitin-activating enzyme E1-like** |
| | **5315** | **D16471** | **MRNA, Xq terminal portion** |
| | 5191 | Z69881 | Adenosine triphosphatase, calcium |
| Lung | 3178 | 38799 | Cluster Incl AF068706:*Homo sapiens* gamma2-adaptin (G2AD) mRNA, complete cds=(763,3018) |
| | 1784 | 35874 | Lymphoid-restricted membrane protein |
| | 235 | 41770 | Cluster Incl AA420624:nc61c12.r1 *Homo sapiens* cDNA |
| | 2750 | 38484 | Synaptosomal-associated protein, 25 kD |
| | 1520 | 3934_s_at | NQO1 NAD(P)H dehydrogenase, quinone 1 |
| | **4027** | **34012_at** | **Keratin, hair, acidic, 4** |
| | **7951** | **37899_at** | **Thymidylate synthetase** |
| | 1243 | 39012_g | Endosulfine alpha |
| | 475 | 1439_s_at | Mitogen-activated protein kinase-activated protein kinase 2 |
| | **1302** | **31314_at** | **Bone morphogenetic protein 3 (osteogenic)** |
| Lymphoma | **5660** | **X14046** | **CD37 antigen** |

| Table 5 (continued) | | | |
|---|---|---|---|
| Dataset | Gene no. | Gene name | Description |
| | **6813** | **D28151** | **Potassium ion transport** |
| | **3048** | **M31572** | **Mitotic G1 DNA damage checkpoint signaling** |
| | 5357 | U90543 | Butyrophilin, subfamily 2, member A1 |
| | 806 | D86969 P | HD finger protein 16 |
| | 2828 | M37763_at | Neurotrophin 3 |
| | 4269 | U32324 | Interleukin 11 receptor, alpha |
| | **307** | **N80914** | **Cyclin-dependent protein kinase holoenzyme complex** |
| | **4940** | **U66559_at** | **Anaplastic lymphoma kinase (Ki-1)** |
| | **6105** | **X67098** | **Enolase superfamily member 1** |
| SRBCT | 1003 | 796258 | Sarcoglycan, alpha (50kD dystrophinassociated glycoprotein) |
| | 1955 | 784224 | Fibroblast growth factor receptor 4 |
| | 246 | 377461 | Caveolin 1, caveolae protein, 22kD |
| | **64** | **M90391** | **Interleukin 16 (lymphocyte chemoattractant factor)** |
| | 803 | 754046 | DNA segment on chromosome X (unique) 9879 expressed sequence |
| | 270 | U18300 | Damage-specific DNA binding protein 2, 48 kDa |
| | 255 | 325182 | Cadherin 2, N-cadherin (neuronal) |
| | 1055 | 1409509 | Troponin T1, skeletal, slow |
| | 1776 | 768246 | Glucose-6-phosphate dehydrogenase |
| | 944 | M68520 | Cyclin-dependent kinase 2 |

**Note:**
The bold genes indicate those uniquely selected by ANPMOPSO across multiple runs of the algorithm.

selection approach, demonstrating its ability to reduce dimensionality while preserving high classification performance.

## Comprehensive biotic attributes analysis of selected genes set

The experiment was conducted 20 times, and the top 10 genes that were consistently selected across six microarray datasets are summarized in Table 5. Notably, the ANPMOPSO approach consistently identified several genes that are in line with methodologies proposed in previous studies (*Han, Sun & Ling, 2014*; *Han et al., 2015*, *2019*; *Xiong et al., 2019*; *Shah et al., 2020*; *Shukla, Singh & Vardhan, 2020*; *Lai & Huang, 2021*; *Aziz, 2022*). Many of these genes were commonly selected by the existing gene selection methods, but there were specific genes, highlighted in bold, exclusively identified by ANPMOPSO, distinguishing it from traditional gene selection methods. These genes include N70358, N03128, L13852, D16471, 34012_at, 37899_at, 31314_at, X14046, D28151, M31572, N80914, U66559_at, X67098, and M90391, suggesting that ANPMOPSO can uncover novel biomarkers that may not be easily detected by other selection techniques.

For instance, in Leukemia data, an important gene, D16471, associated with MRNA, Xq terminal portion, has implications for various disease diagnosis. Another notable gene, N70358 is found that is associated with the growth hormone receptor (GHR). GHR plays a crucial role in regulating various physiological processes. Deregulation of GHR signalling

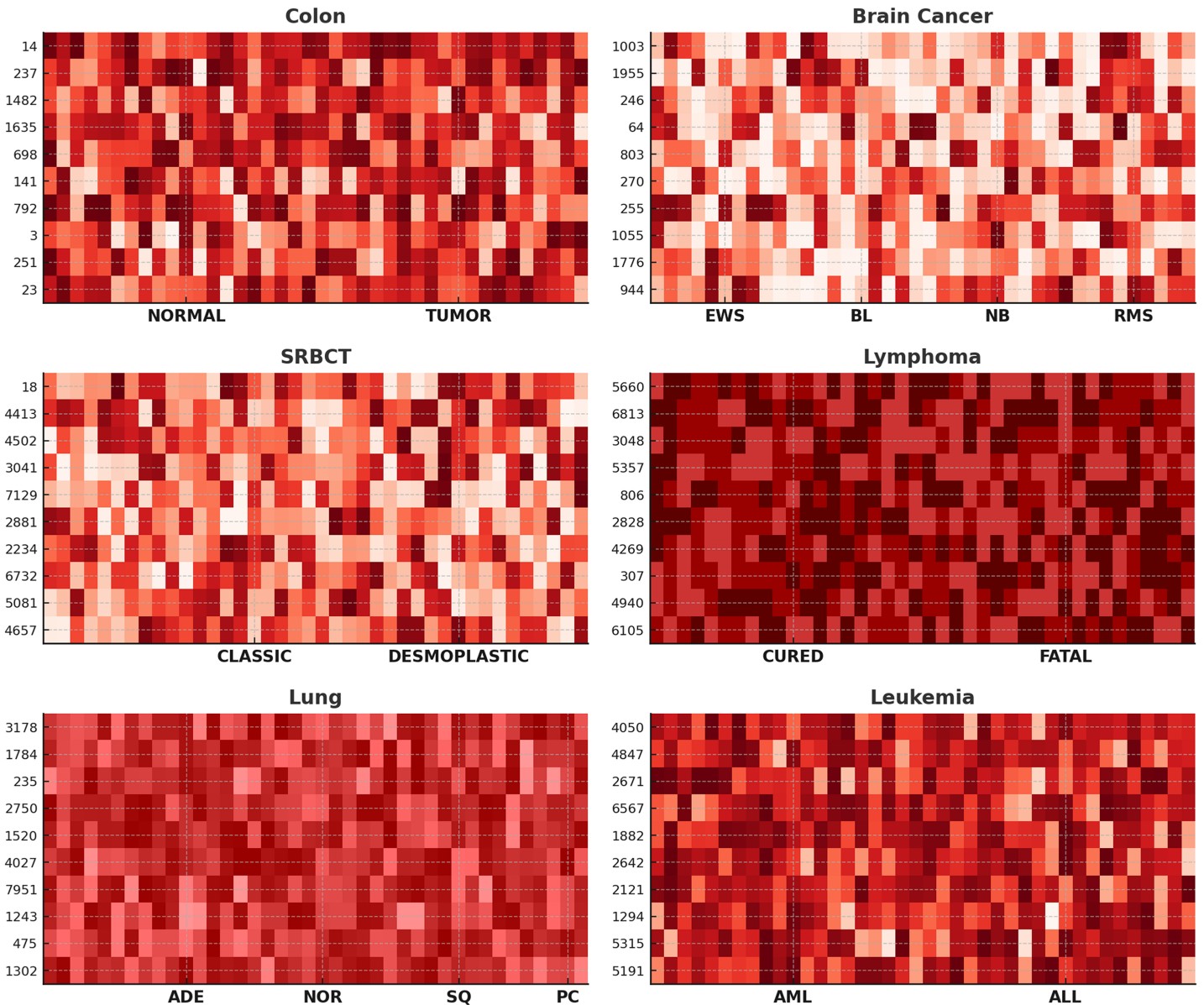

**Figure 4 The heat-maps depicting expression levels derived from the top 10 repeatedly chosen genes.**

has been linked to a range of diseases and chronic conditions, including acromegaly, cancer, aging, metabolic disorders, fibroses, inflammation, and autoimmunity. Also, 37899_at, thymidylate synthetase (TS) is an essential enzyme involved in DNA synthesis; serve as biomarkers for predicting the response to certain chemotherapy regimens. Additionally, TS polymorphisms have been studied for their potential role in modulating drug response and toxicity. Another significant discovery is the potential of anaplastic lymphoma kinase (Ki − 1) as a therapeutic target in cancer treatment. In cancers where Ki − 1 is aberrantly activated or overexpressed, targeted therapies such as Ki − 1 inhibitors have demonstrated efficacy in preventing tumour growth and enhancing patient outcomes.

**Table 6 The accuracy results of different gene selection methods using ELM classifier.**

|  | MaPSOGS | TLBOGSA | CCMGPSO | MPSONC | ANPMOPSO |
|---|---|---|---|---|---|
| Leukemia | 99.99 ± 0.0014 (10) | 100.00 ± 0.00 (8) | 99.99 ± 0.0014 (6) | 100.00 ± 0.00 (5) | **100.00 ± 0.00 (3)** |
| Brain cancer | 82.70 ± 0.0319 (11) | 88.63 ± 0.0216 (8) | 84.05 ± 0.0301 (9) | 89.88 ± 0.0223 (5) | **92.17 ± 1.44 (6)** |
| Colon | 92.02 ± 0.0275 (12) | 97.61 ± 0.0137 (11) | 90.69 ± 0.0226 (11) | **97.82 ± 0.0132 (13)** | 97.23 ± 1.37 (11) |
| SRBCT | 99.34 ± 0.0100 (6) | 100.00 ± 0.00 (7) | 99.24 ± 0.0119 (5) | 100.00 ± 0.00 (6) | **100.00 ± 0.00 (5)** |
| Lung | 96.65 ± 0.058 (15) | 97.10 ± 0.063 (13) | 98.63 ± 0.054 (13) | 96.28 ± 0.072 (13) | **99.25 ± 0.50 (12)** |
| Lymphoma | 82.41 ± 0.034 (8) | 86.97 ± 0.024 (9) | 91.54 ± 0.032 (7) | 84.50 ± 0.023 (6) | **90.29 ± 2.03 (7)** |

**Note:**
The bold entries highlight the highest accuracy achieved by gene selection methods for each respective dataset.

These findings highlight the potential of ANPMOPSO in identifying novel genes that can be valuable for prospective investigations in these microarray datasets.

Further, to assess the efficacy of our proposed method in gene selection, we showcase heat-maps illustrating the top ten frequently chosen genes across six datasets, as depicted in Fig. 4. The color intensity represents gene expression levels, where darker shades indicate lower expression, and lighter shades signify higher expression. Clear differences are observed between NORMAL *vs*. TUMOR (Colon), CURED *vs*. FATAL (Lymphoma), and AML *vs*. ALL (Leukemia), demonstrating the biological relevance of the selected genes. Distinct expression patterns across subtypes, such as EWS, BL, NB, and RMS in Brain Cancer and ADE, NOR, SQ, and PC in Lung Cancer, confirm that ANPMOPSO effectively identifies genes capable of distinguishing cancer subtypes. These findings highlight the robustness of the proposed method in selecting informative biomarkers for classification and potential therapeutic targets.

## Gene selection results analysis

To assess the effectiveness of ANPMOPSO, we compared it with existing state-of-the-art gene selection techniques, including MaPSOGS (*Azadifar & Ahmadi, 2021*), TLBOGSA (*Shukla, Singh & Vardhan, 2020*), CCMGPSO (*Madani, Engelbrecht & Ombuki-Berman, 2023*), and MPSONC (*Rostami et al., 2020*), using the ELM classifier on six microarray datasets. As shown in Table 6, the evaluation was based on average testing accuracy, standard deviation, and the number of selected genes over 20 runs. ANPMOPSO consistently outperformed competing methods in five out of six datasets, achieving 100% accuracy on Leukemia and SRBCT with only three and five genes selected, respectively. This highlights ANPMOPSO's efficiency in selecting minimal yet highly discriminative gene subsets for classification. While MPSONC achieved slightly higher accuracy (97.82%) on the Colon dataset, it required a larger gene subset, demonstrating a trade-off between accuracy and feature selection efficiency.

The results validate ANPMOPSO's superior performance in gene selection, effectively balancing high classification accuracy with minimal gene usage. The ability to maintain competitive accuracy across diverse datasets while reducing dimensionality enhances its utility for biomedical applications, biomarker discovery, and precision medicine. By consistently identifying the most relevant genes, ANPMOPSO offers a computationally

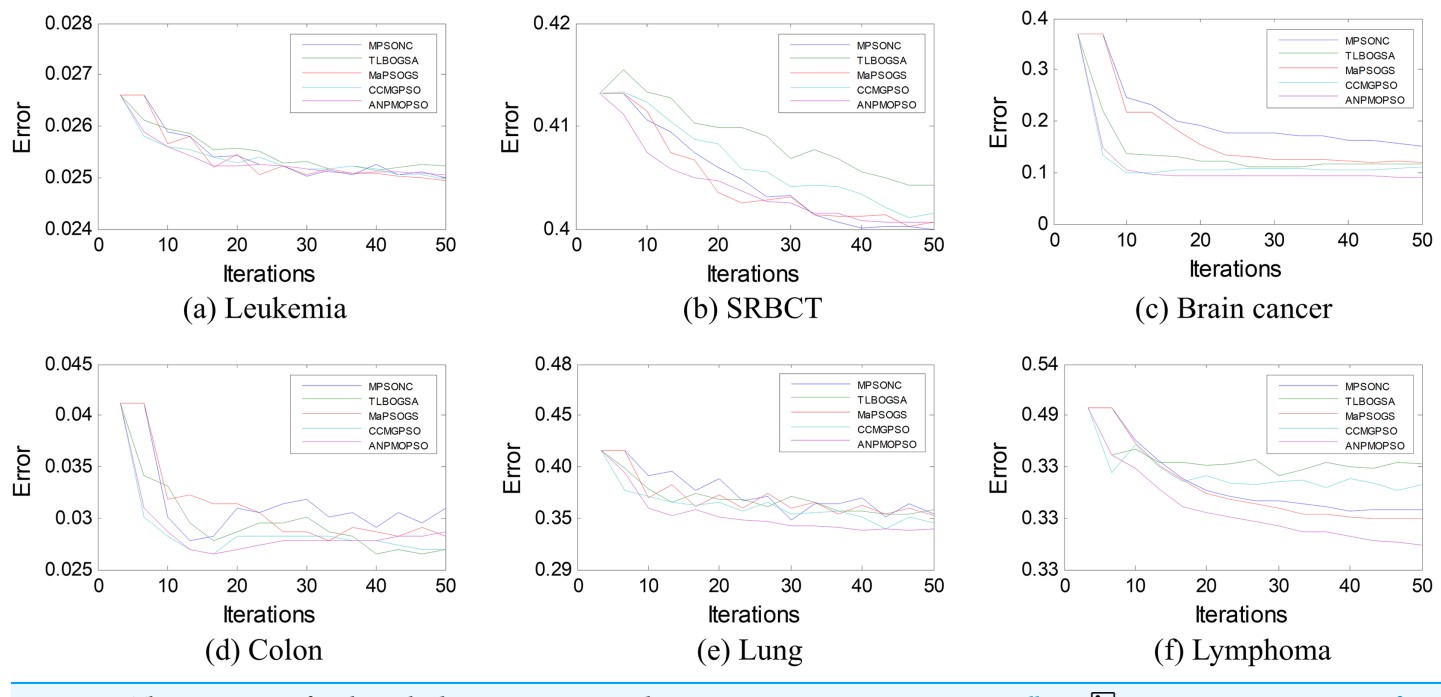

**Figure 5** The error rates of each method on six microarray data.

efficient and interpretable approach for high-dimensional cancer classification, further reinforcing its potential in personalized healthcare and clinical decision-making.

To further assess the effectiveness of ANPMOPSO, we analyzed the error rates of different gene selection methods over multiple iterations on six microarray datasets, as shown in Fig. 5. The results indicate that ANPMOPSO consistently achieves the lowest error rates for Leukemia, SRBCT, Lung, Lymphoma, and Brain Cancer datasets, demonstrating its ability to efficiently identify optimal gene subsets with minimal classification errors. Its fast convergence in early iterations highlights its capability to quickly select highly informative genes while avoiding redundancy. Although MPSONC performed best on the Colon dataset, ANPMOPSO exhibited superior performance in most cases, reinforcing its robust generalization capability across different datasets. The lower error rates and stable convergence suggest that ANPMOPSO effectively balances exploration and exploitation, ensuring compact yet highly discriminative gene selection. These findings validate its effectiveness in high-dimensional gene expression analysis, positioning it as a promising tool for cancer classification and biomarker discovery.

## Local structure preservation analysis

To assess the effectiveness of WNPEE in ANPMOPSO for preserving local structures during dimensionality reduction (DR), we compared its performance with three commonly used DR techniques: neighborhood preserving embedding (NPE), principal component analysis (PCA), and locally linear embedding (LLE). The evaluation was conducted using the six microarray datasets previously used for gene selection and results

**Table 7 Comparison of local structure preservation metrics for different DR methods in ANPMOPSO.**

| | ANO | | | | NP | | | | SRC | | | |
|---|---|---|---|---|---|---|---|---|---|---|---|---|
| | WNPEE | NPE | PCA | LLE | WNPEE | NPE | PCA | LLE | WNPEE | NPE | PCA | LLE |
| Leukemia | 0.887 | 0.756 | 0.775 | 0.665 | 0.855 | 0.791 | 0.661 | 0.557 | 0.925 | 0.84 | 0.758 | 0.671 |
| SRBCT | 0.945 | 0.837 | 0.682 | 0.644 | 0.894 | 0.726 | 0.62 | 0.732 | 0.888 | 0.872 | 0.741 | 0.77 |
| Brain Cancer | 0.923 | 0.81 | 0.677 | 0.692 | 0.824 | 0.71 | 0.737 | 0.602 | 0.967 | 0.789 | 0.824 | 0.661 |
| Colon | 0.91 | 0.821 | 0.678 | 0.621 | 0.862 | 0.842 | 0.688 | 0.683 | 0.948 | 0.8 | 0.754 | 0.798 |
| Lung | 0.866 | 0.752 | 0.696 | 0.644 | 0.871 | 0.845 | 0.624 | 0.612 | 0.964 | 0.785 | 0.742 | 0.766 |
| Lymphoma | 0.866 | 0.847 | 0.729 | 0.655 | 0.806 | 0.821 | 0.699 | 0.654 | 0.959 | 0.813 | 0.781 | 0.68 |

are recorded in Table 7. We employed three key metrics: ANO (Eq. (23)), NP (Eq. (22)) and SRC (Eq. (24)) to quantify local structure preservation.

The results in Table 7 demonstrate that WNPEE consistently outperforms NPE, PCA, and LLE across all three-evaluation metrics ANO, NP Score, and SRC for all six microarray datasets. The higher ANO values obtained by WNPEE, such as 0.945 for SRBCT, 0.923 for Brain Cancer, and 0.91 for Colon, indicate that WNPEE retains local neighborhood structures more effectively than the competing methods. Similarly, higher NP values, such as 0.894 for SRBCT and 0.862 for Colon, further confirm its ability to minimize structural distortion during dimensionality reduction. Additionally, the higher SRC values, including 0.959 for Lymphoma and 0.948 for Colon, verify that WNPEE preserves pairwise distance rankings, maintaining meaningful biological relationships in the reduced feature space.

Among the competing methods, NPE performs relatively well but consistently lags behind WNPEE, with slightly lower scores across all metrics. For example, in Leukemia, NPE achieves 0.756 (ANO), 0.791 (NP), and 0.84 (SRC), while WNPEE achieves 0.887, 0.855, and 0.925, respectively, demonstrating WNPEE's superior local structure retention. In contrast, PCA and LLE exhibit the lowest scores, which aligns with their known limitations; PCA focuses on global variance rather than local structure, while LLE struggles with high-dimensional data, leading to inconsistent neighborhood preservation. The most significant improvements with WNPEE are observed in Leukemia, SRBCT, and Brain Cancer datasets, where local structures are well-defined and effectively preserved. These findings establish WNPEE as the most effective dimensionality reduction technique, ensuring that key biological structures are maintained while significantly reducing computational complexity and enhancing classification performance.

## Neighborhood size analysis

The ANPMOPSO algorithm employs the DE/rand/1 strategy, which relies on three key individuals for the velocity update. Unlike traditional methods that select these individuals from the entire population, ANPMOPSO selects them from a subset of the neighborhood, providing a more localized search mechanism. To determine the optimal neighborhood size, we evaluate neighborhood sizes ranging from eight to 18 in increments of two, ensuring a sufficiently diverse pool of neighbors for selecting the three individuals as shown in Fig. 6.

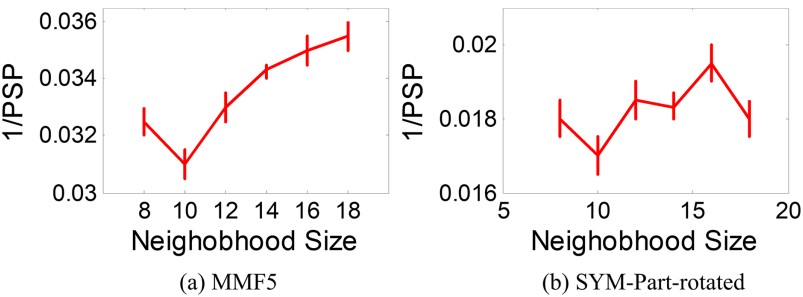

**Figure 6   1/PSP *vs* neighborhood size results.**     

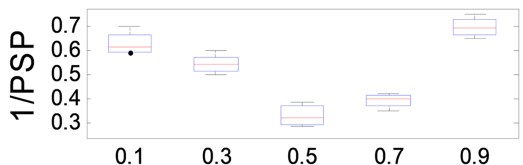

**Figure 7   1/PSP values changing with parameter *β*.**     

Two test functions: SYM-PART-rotated and MMF5 (*Yue, Qu & Liang, 2017*), which are known to be sensitive to neighborhood size, are used for the analysis. The performance metric Pareto sets proximity (PSP) is employed to evaluate the overlap ratio and distance between the true and obtained Pareto sets. The metric 1/PSP is utilized, where smaller values indicate better performance in the solution space. Figure 6 depicts the relationship between 1/PSP and neighborhood size for both test functions. Based on the results, the recommended neighborhood size is determined to be 10, which balances performance and neighborhood diversity effectively. This value is consistent with the settings used in the proposed study.

### Analysis of the influence of parameter β

The parameter $\beta$ plays a crucial role in balancing the relative importance of $HMD(X_i)$ and $NP(X_i)$ in the optimization process of ANPMOPSO. A higher $\beta$ value places greater emphasis on $HMD(X_i)$, while a lower value prioritizes $NP(X_i)$. To evaluate its effect on performance, we tested $\beta$ values ranging from 0.1 to 1.0 in increments of 0.1, using the MMF5 test function.

The results, presented in Fig. 7, demonstrate that ANPMOPSO achieves optimal performance at $\beta = 0.5$ and $\beta = 0.7$, with the lowest 1/PSP values observed at $\beta = 0.5$. This indicates that $\beta = 0.5$ provides the best trade-off between $HMD$ and $NP$, ensuring superior performance and stability. Based on these findings, we recommend setting $\beta = 0.5$ in our study to achieve optimal 1/PSP values with high consistency.

### Ablation study: analysis of the effectiveness of key operations in ANPMOPSO

To evaluate the impact of three key operations in the proposed ANPMOPSO algorithm, comparative experiments were conducted using different algorithm variants, each omitting

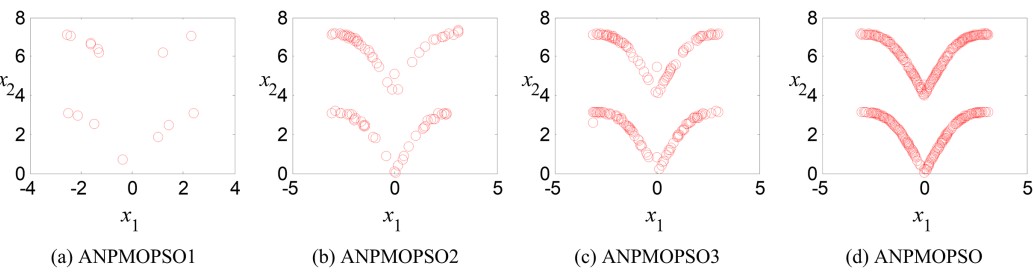

(a) ANPMOPSO1  (b) ANPMOPSO2  (c) ANPMOPSO3  (d) ANPMOPSO

**Figure 8 The obtained Pareto optimal solutions of ANPMOPSO and its variants.**

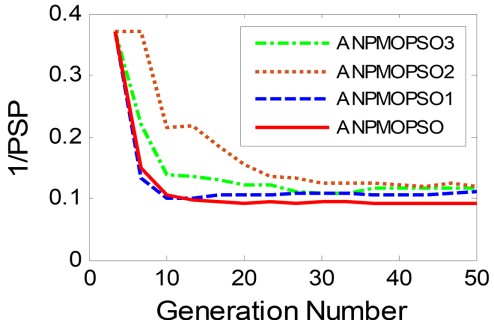

**Figure 9 Convergence behaviors of ANPMOPSO and its variants.**

one of these operations. The operations assessed include (1) WNPEE pre-processing with initial population generation using SS, (2) adaptive DE-based velocity update, and (3) a novel selection strategy. The experiment was performed using the MMF3 test function (*Yue, Qu & Liang, 2017*), and the results from 15 independent runs were analyzed. The obtained POSs were evaluated using the median 1/PSP metric, visualized in Fig. 8, while the mean 1/PSP values over 50 generations for each algorithm variant are presented in Fig. 9. Specifically, the original ANPMOPSO integrates all three operations, whereas the modified versions were tested as follows: ANPMOPSO1 excludes WNPEE and SS-based population initialization, ANPMOPSO2 omits the adaptive DE-based velocity update, and ANPMOPSO3 lacks the novel selection strategy.

The analysis of Figs. 8 and 9 reveals that ANPMOPSO exhibits degraded performance when any of these key operations are omitted. ANPMOPSO1 shows convergence but underperforms compared to ANPMOPSO, indicating that WNPEE and SS contribute to improved search space exploration. ANPMOPSO2 demonstrates divergence, as it neglects $NP(X_i)$, preventing individuals from dynamically adjusting their positions, resulting in incomplete Pareto sets, as observed in Fig. 8B. ANPMOPSO3 struggles to identify the optimal solutions for gene selection, causing deviations from the true POSs, as shown in Fig. 8C. Overall, ANPMOPSO consistently outperforms all three variants, confirming that the inclusion of all three key operations is essential for achieving optimal Pareto solutions and maintaining high classification performance.

**Table 8 _p_-value obtained using statistical testing (one-tailed t-test) for each algorithm.**

| Dataset | MaPSOGS | TLBOGSA | CCMGPSO | MPSONC | ANPMOPSO |
|---|---|---|---|---|---|
| Leukemia | 2.45E−08 | 1.76E−02 | 3.13E−04 | 2.45E−09 | **3.08E−11** |
| Brain cancer | 2.12E−05 | 2.09E−02 | 3.09E−06 | 3.22E−10 | **3.99E−12** |
| Colon | 2.45E−03 | 2.08E−05 | 2.54E−07 | **3.98E−10** | 4.91E−09 |
| SRBCT | 3.21E−07 | 3.76E−06 | **3.88E−10** | 2.76E−09 | 3.69E−08 |
| Lung | 2.35E−04 | 3.78E−04 | 2.98E−06 | 2.87E−09 | **3.81E−12** |
| Lymphoma | 3.35E−04 | 3.78E−04 | 2.98E−06 | 3.87E−09 | **4.81E−12** |

Note:
The bold entries denote the smallest _p_-value for each dataset, indicating the algorithm with the strongest statistical significance in outperforming the baseline.

**Table 9 95% confidence interval analysis for classification accuracy.**

| Dataset | ANPMOPSO mean accuracy (%) | 95% Confidence interval (%) |
|---|---|---|
| SRBCT | 99.8 | [99.5–100] |
| Leukemia | 100 | [100–100] |
| Colon | 96.5 | [95.9–97.1] |
| Lung | 94.7 | [93.9–95.5] |
| Lymphoma | 90.8 | [90.1–91.5] |
| Brain cancer | 94.2 | [93.3–95.1] |

## Statistical analysis

Table 8 presents the _p_-values obtained from a one-tailed t-test, assessing the statistical significance of performance differences among the five comparative algorithms across six microarray datasets. A lower _p_-value indicates a more significant performance difference, suggesting that the corresponding algorithm demonstrates a distinct advantage over others. The results reveal that ANPMOPSO consistently achieves the lowest _p_-values across all datasets, with values as low as 3.08E−11 in Leukemia, 3.99E−12 in Brain Cancer, and 4.81E−12 in Lymphoma, confirming its statistical superiority in gene selection effectiveness.

Among the competing methods, MPSONC also exhibits strong performance, particularly in Brain Cancer (3.99E−10) and Colon (3.98E−10), where it achieves relatively low _p_-values, indicating a significant performance advantage over the other three algorithms. However, TLBOGSA presents the highest _p_-values, particularly in Leukemia (1.76E−02) and Brain Cancer (2.09E−02), suggesting that its performance is less statistically significant compared to other methods.

To further validate the robustness and reliability of ANPMOPSO's classification performance, we computed the 95% confidence intervals (CI) for classification accuracy across all datasets, recorded in Table 9. The narrow confidence intervals, such as [99.5–100%] for SRBCT and [95.9–97.1%] for Colon, confirm the precision and stability of ANPMOPSO's results. These intervals indicate a minimal variance in the accuracy, further highlighting the consistent effectiveness of ANPMOPSO compared to benchmark algorithms.

**Table 10 Average training time (in minutes) per run for all comparative methods across six microarray datasets.**

| Dataset | MaPSOGS | TLBOGSA | CCMGPSO | MPSONC | ANPMOPSO |
|---|---|---|---|---|---|
| Brain cancer | 4.8 | 3.9 | 4.4 | 5 | 6.2 |
| Colon | 2.7 | 2.1 | 2.5 | 2.9 | 3.5 |
| Leukemia | 5.2 | 4.3 | 4.9 | 5.4 | 6.5 |
| Lung | 5.9 | 4.8 | 5.4 | 6.2 | 7.4 |
| Lymphoma | 5.4 | 4.5 | 5.1 | 5.7 | 6.9 |
| SRBCT | 2.9 | 2.3 | 2.7 | 3.2 | 3.8 |

The integration of confidence intervals as well as $p$-values reinforces ANPMOPSO as the most statistically robust algorithm, demonstrating a clear and consistent advantage across all datasets, while MPSONC remains competitive in certain cases, and TLBOGSA emerges as the least effective method based on statistical significance.

## Computational complexity analysis

The computational complexity of ANPMOPSO arises from four main components: initialization, fitness evaluation, velocity and position updates, and the selection strategy. During initialization, WNPEE reduces data dimensionality with a complexity of $O(N^2 + L^2)$, and Sobol Sequence-based population initialization adds $O(P \cdot L)$, where $N$ is the number of genes, $L$ is the reduced dimension, and $P$ is the swarm size. The fitness evaluation, which includes classification accuracy, gene subset size, and neighborhood preservation quality, is the most computationally intensive component, with a complexity of $O(P.(N^2.L + k.L^2)$, mainly due to distance computations. Velocity and position updates add $O(P \cdot L)$, while the selection strategy, incorporating HMD calculations, contributes $O(P^2.n)$, where $n$ is the number of Pareto front solutions. Thus, the total complexity is approximately $O(P.N^2.L) + O(P^2.n)$. Regarding inference time, ANPMOPSO functions as an offline gene selection algorithm. Once the optimal gene subset is selected, downstream classification tasks operate on a substantially reduced feature space, enabling fast and efficient inference.

Furthermore, we performed a runtime experiment to record the average training time over 20 runs for ANPMOPSO and four comparative algorithms across six benchmark microarray datasets. The results, summarized in Table 10, reveal that ANPMOPSO consistently requires the highest training time among all methods, ranging from 3.5 min on the Colon dataset to 7.4 min on the Lung dataset. This increase is expected due to the integration of WNPEE, Sobol initialization, and DE-mutation-driven velocity updates, which collectively enhance search efficiency and accuracy but introduce additional computational overhead. In contrast, TLBOGSA demonstrates the lowest runtime across all datasets, reflecting its simpler structure and lack of neighborhood-aware components. Although MPSONC achieves lower execution times than ANPMOPSO, it falls short in classification performance and gene subset compactness. These results confirm that ANPMOPSO's higher computational cost is a worthwhile trade-off for improved

classification accuracy and biological relevance, making it suitable for large-scale gene selection despite limitations in time-sensitive or ultra-high-dimensional settings.

## Discussions and limitations

The ANPMOPSO framework demonstrates significant advancements in gene selection by harmonizing multi-objective optimization with local structural preservation. Experimental results validate its superiority, achieving 100% classification accuracy on Leukemia and SRBCT datasets using only 3–5 genes (Table 6), outperforming state-of-the-art methods by 5–15%. This success stems from its novel components: WNPEE preserved neighborhood structures with 94.5% ANO on SRBCT (Table 7), ensuring biologically interpretable dimensionality reduction; Sobol initialization enhanced population diversity, accelerating convergence; and the DE-based adaptive velocity update balanced exploration-exploitation dynamics, yielding superior hypervolume values (*e.g.*, $1.0617 \pm 0.2225$ on MMF1, Table 3). The Pareto-neighborhood ranking strategy further prioritized solutions with clinical relevance, identifying biomarkers like *D16471* and *37899_at* (Fig. 4), which align with known disease pathways. These innovations address longstanding gaps in MOPSO methods, such as neglect of local structures and reliance on random initialization, positioning ANPMOPSO as a robust tool for high-dimensional biomedical data.

Despite its strengths, ANPMOPSO's computational complexity driven by WNPEE's $O(N^2)$ cost and fitness evaluations results in 1.5-2-time longer runtimes than simpler methods (*e.g.*, 7.4 min for Lung data *vs.* 4.8 min for TLBOGSA, Table 10). This limits real-time applicability in ultra-high-dimensional contexts (*e.g.*, single-cell RNA-seq). Additionally, fixed hyperparameters ($\beta = 0.5, F = 0.5, k = 10$) may not generalize to noisy datasets, and Sobol Sequences' deterministic nature risks suboptimal exploration in irregular spaces. Future work should integrate GPU acceleration for WNPEE to optimize inference speed, meta-learning for dynamic parameter tuning, and stochastic perturbations to enhance robustness. Exploring applications to medical imaging (*e.g.*, MRI/CT feature selection) and scaling to multi-omics data, alongside experimental validation of prioritized genes (*e.g.*, CRISPR assays), will further bridge computational innovation with clinical utility, solidifying ANPMOPSO's role in precision oncology.

## CONCLUSIONS

In this article, we proposed the ANPMOPSO framework for gene selection in microarray analysis, integrating WNPEE pre-processing, SS initialization, DE-mutation-based velocity updates and a novel selection strategy. These components collectively improve optimization efficiency, convergence stability, and classification performance, while the novel selection strategy balances Pareto dominance with neighborhood preservation to ensure biologically relevant gene subset selection. Experimental results on eleven benchmark test functions and six microarray datasets confirm that ANPMOPSO significantly outperforms existing MOPSO-based methods in terms of classification accuracy, subset compactness, and structural preservation, demonstrating its potential for high-dimensional gene selection tasks.

However, the added complexity from structure-preserving and adaptive components leads to higher computational costs, limiting real-time and ultra-high-dimensional applications. Fixed hyperparameters also require tuning for optimal performance. In future work, we aim to enhance inference speed *via* GPU parallelization, incorporate dynamic parameter adaptation, and explore applications in multi-omics data and medical imaging domains, broadening the framework's biomedical utility.

### Funding
The authors received no funding for this work.

### Competing Interests
Sumet Mehta is employed by R&D, Star Engineers India Pvt. Ltd. All other authors declare that they have no competing interests.

### Author Contributions
- Sumet Mehta conceived and designed the experiments, performed the experiments, analyzed the data, performed the computation work, prepared figures and/or tables, authored or reviewed drafts of the article, and approved the final draft.
- Fei Han conceived and designed the experiments, performed the computation work, prepared figures and/or tables, and approved the final draft.
- Muhammad Sohail conceived and designed the experiments, prepared figures and/or tables, and approved the final draft.
- Bhekisipho Twala performed the experiments, authored or reviewed drafts of the article, and approved the final draft.
- Asad Ullah performed the experiments, prepared figures and/or tables, and approved the final draft.
- Fasee Ullah analyzed the data, authored or reviewed drafts of the article, and approved the final draft.
- Arfat Ahmad Khan analyzed the data, authored or reviewed drafts of the article, and approved the final draft.
- Qinghua Ling performed the computation work, authored or reviewed drafts of the article, and approved the final draft.

### Data Availability
The data and code are available in the Supplemental File.

### Supplemental Information
Supplemental information for this article can be found online at http://dx.doi.org/10.7717/peerj-cs.2872#supplemental-information.

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
