# Peer review of "Gene selection based on adaptive neighborhood-preserving multi-objective particle swarm optimization"

_PeerJ Computer Science, doi:10.7717/peerj-cs.2872_

## Round 0.1 · original submission · Major Revisions

Dear Authors,

Thank you for submitting your article. Feedback from the reviewers is now available. It is not recommended that your article be published in its current format. However, we strongly recommend that you address the issues raised by the reviewers, especially those related to readability, methodological transparency and ethical considerations, experimental design and validity, and resubmit your paper after making the necessary changes.

Reviewer 2 has asked you to provide specific references. You are welcome to add them if you think they are relevant. However, you are not obliged to include these citations, and if you do not, it will not affect my decision.

Best wishes,

Reviewer 1 ·

Basic reporting

- Clear and unambiguous, professional English used throughout: The manuscript is well-structured and written in professional English. However, certain sections contain complex and lengthy sentences that could be refined for better readability. For example:
"The proposed framework incorporates several key innovations: (1) Weighted Neighborhood-Preserving Ensemble Embedding (WNPEE) for dimensionality reduction while retaining essential local data structures, (2) Sobol Sequence (SS)-based initialization to enhance population diversity and mitigate sensitivity to initialization, (3) an adaptive velocity update mechanism with Differential Evolution (DE) mutation to balance exploration and exploitation, and (4) a novel ranking mechanism that integrates Pareto dominance with neighborhood preservation quality to ensure the selection of biologically meaningful and interpretable gene subsets."
This sentence is overly complex and could be rewritten in a more concise manner to improve clarity.
- Literature references, sufficient field background/context provided: The manuscript provides a comprehensive background on gene selection and optimization methodologies. Relevant prior work is cited appropriately, but a deeper comparative discussion on how this study differs from previous approaches would strengthen the justification for the research. The introduction successfully contextualizes the importance of the research, but a clearer statement of the research gap would be beneficial.
.
- Professional article structure, figures, tables. Raw data shared: The structure follows the standard format, including sections for methodology, experimental results, and discussion. Figures and tables are relevant and well-labeled, though some captions lack explanatory details. For example, Figure 2 could include a more detailed description of how the Pareto-optimal solutions were generated.
- Self-contained with relevant results to hypotheses: The study is well-contained and presents all relevant findings in alignment with the stated objectives. However, the hypothesis could be stated more explicitly to align better with the reported findings. In the abstract section the author should emphasize the priority of the ANPMOPSO method by giving some superior numeric performance metrics.

Experimental design

- Original primary research within Aims and Scope of the journal: The study aligns well with PeerJ Computer Science’s scope, offering an innovative approach to gene selection using multi-objective optimization. The problem statement is well-defined and relevant to bioinformatics research.
- Research question well defined, relevant & meaningful. It is stated how research fills an identified knowledge gap: The research question is clear, addressing the challenge of selecting biologically significant gene subsets while optimizing classification accuracy, feature subset size, and neighborhood preservation. While the manuscript outlines the gap in existing MOPSO-based methods, it would benefit from a more detailed discussion on why existing methods fail and how the proposed method explicitly addresses these limitations.
- Rigorous investigation performed to a high technical & ethical standard: The methodology is rigorous, employing a combination of Weighted Neighborhood-Preserving Ensemble Embedding (WNPEE), Sobol Sequence-based initialization, and Differential Evolution-based optimization. Ethical considerations are not applicable as the study does not involve human or animal subjects.
- Methods described with sufficient detail & information to replicate: The methods are described with clarity, but additional implementation details would improve reproducibility. Key hyperparameters such as particle swarm size, mutation rates, and initialization settings should be explicitly stated. A flowchart illustrating the step-by-step execution of the ANPMOPSO framework would improve accessibility. And detailed ablation study should be performed by employing PSO method and using other modules separately. The contribution of each module should be shown experimentally.

Validity of the findings

- Impact and novelty not assessed. Meaningful replication encouraged where rationale & benefit to literature is clearly stated: The paper does not overemphasize impact but provides comparisons with existing methods to demonstrate improvements. The authors clearly explain the rationale for using replication techniques to validate the robustness of their findings.
- All underlying data have been provided; they are robust, statistically sound, & controlled: The manuscript presents classification accuracy, F1-scores, and Pareto-optimal solution comparisons across benchmark datasets. However, additional statistical significance tests (e.g., p-values, confidence intervals) would further support the robustness of the reported improvements.
- Conclusions are well stated, linked to original research question & limited to supporting results: The conclusions align with the experimental findings and do not overstate claims. Some statements (e.g., "establishes ANPMOPSO as a powerful and effective tool for addressing high-dimensional gene selection challenges") should be more cautiously framed given that only six datasets were tested.

Additional comments

This review critically evaluates the article titled "Gene Selection Based on Adaptive Neighborhood-Preserving Multi-Objective Particle Swarm Optimization." The study focuses on multi-objective optimization techniques to enhance gene selection and classification accuracy while preserving neighborhood structures. The primary goal is to improve feature selection, classification performance, and biological interpretability using adaptive particle swarm optimization techniques. However, in its current form, the manuscript presents several drawbacks that require attention. This review assesses the novelty, technical soundness, contribution to the field, and overall relevance of the work. The suggested revisions outlined below are expected to enhance the clarity, rigor, and impact of the article.
1. The structure follows the standard format, but a separate section titled “Dataset” should be added to provide a detailed description of the dataset properties.
2. The “Discussions" section should be added in a more highlighting, argumentative way. The evaluation results should be described in more details including the discussion about the algorithm complexity. The discussion section could benefit from an exploration of potential limitations, such as computational demands or limitations in real-time processing.
3. The limitations of the algorithm and the evaluation environment should be discussed in the paper. What are the capabilities, benefits and limitations of study? Suggestions for future research, like optimizing the model for faster inference or exploring its application to other types of medical imaging, would add depth to the study.
4. The complexity of the proposed model and the model parameter uncertainty are not enough mentioned.
5. As in the abstract section, to prove your claim about computational efficiency, you should calculate the computational efficiency of the model, including training time and inference time.
6. The article lacks comparisons with state-of-the-art methods, specifically those developed after 2024. The authors should compare the results of their method with those of previous studies. As mentioned in the literature, there are several methods with very high accuracy, even better than the proposed method. Author(s) can do compare table (A new table can add about previous studies to result section.).
7. How did you set the parameters of proposed method for better performance?
8. Explain the performance metrics in a subsection.
9. The authors need to show clearly what their contribution is to the body of knowledge.
10. The performance of the proposed method should be better analyzed, commented, and visualized in the experimental section.
11. Improve writing clarity by simplifying complex sentences.
12. Discuss limitations more explicitly, especially regarding proposed ANPMOPSO model contributions and difference from other conventional optimization methods.

Reviewer 2 ·

Basic reporting

This paper proposes a novel ANPMOPSO framework that integrates weighted neighbor-preserving clustering embeddings, Sobol sequence initialization, and adaptive velocity update mechanisms to address the issues of sensitivity and local structure preservation in high-dimensional gene selection. This paper has a clear overall structure. However, the manuscript still has room for improvement in terms of clarity, depth, and impact.

Experimental design

1. From the error rate curves of the five methods, it can be observed that each iteration records the current error rate. It is recommended to plot the error rate curves using the global optimal solutions at each iteration, as this will provide a clearer comparison of algorithm performance. Additionally, in the experiment, it would be beneficial to increase the total number of iterations to ensure that all algorithms reach a state of convergence.

Validity of the findings

no comment

Additional comments

2. The introduction is logically coherent, however, some of the references are somewhat outdated and do not adequately highlight the latest developments in the field. The following recent papers are recommended to be appended: (i) Fu Q. et al. MOFS-REPLS: A large-scale multi-objective feature selection algorithm based on real-valued encoding and preference leadership strategy. Information Sciences. (ii) Zhou X, et al. Rough hypervolume-driven feature selection with groupwise intelligent sampling for detecting clinical characterization of lupus nephritis. Artificial Intelligence in Medicine (iii) Li, X., et al., Learning a convolutional neural network for propagation-based stereo image segmentation.
3. Although this work focuses on high-dimensional gene selection, it is advisable to start the introduction by discussing the importance of gene selection from a biomedical perspective. I suggest enhancing the literature review to include recent advancements in cancer data fusion and deep learning architectures. These areas have seen significant progress, with studies demonstrating the potential of integrating multi-omics data and advanced neural networks to improve cancer diagnosis and treatment. For example, (i)Zhao Y. et al. A review of cancer data fusion methods based on deep learning. Information Fusion. (ii) Wang J. et al. Multi-Scale Three-Path Network (MSTP-Net): A new architecture for retinal vessel segmentation. Measurement.
4. It is recommended to include some suitable diagrams when presenting the algorithm, which will contribute to helping readers better understand the concept inside.
5. Some of the references are missing complete publication years and page numbers.
6. The 'POS' abbreviation in line 170 is left unexplained in the manuscript.

---

## Round 0.2 · accepted · Accept

Dear Authors,

Thank you for addressing the comments. Your paper now seems sufficiently improved and ready for publication.

Best wishes,

Reviewer 1 ·

Basic reporting

satisfied

Experimental design

satisfied

Validity of the findings

satisfied

Additional comments

satisfied

Reviewer 2 ·

Basic reporting

no comment

Experimental design

no comment

Validity of the findings

no comment

Additional comments

no comment